# Efficiently Solving Discounted MDPs via Predictions with Unknown Prediction Errors

**Lixing Lyu** [1]   **Jiashuo Jiang** [2]   **Wang Chi Cheung** [3]

## Abstract

We study infinite-horizon discounted Markov decision processes (DMDPs) under a generative model. Motivated by the Algorithms with Advice framework (Mitzenmacher & Vassilvitskii, 2022), we propose a novel framework to investigate how black-box predictions of the transition matrix can enhance sample efficiency in solving DMDPs and improve sample complexity bounds. We focus on DMDPs with $N$ state–action pairs and discount factor $\gamma$. We first provide an impossibility result showing that, in the presence of predictions with unknown accuracy, no sampling policy can compute an $\epsilon$-optimal policy with a sample complexity better than $\tilde{O}((1-\gamma)^{-3}N\epsilon^{-2})$, which matches the state-of-the-art minimax sample complexity bound without prediction. In complement, we design an algorithm based on minimax optimization techniques that leverages predictions of the transition matrix without requiring knowledge of the prediction error. Our algorithm achieves a sample complexity bound that depends on the prediction error and is uniformly better than $\tilde{O}((1-\gamma)^{-4}N\epsilon^{-2})$, the previous best result derived from convex optimization methods. In some cases, our bound even improves upon the state-of-the-art $\tilde{O}((1-\gamma)^{-3}N\epsilon^{-2})$, despite not having access to the prediction quality.

## 1. Introduction

Markov decision processes (MDPs) are a fundamental framework for sequential decision making under uncertainty

---

[1]Institute of Operations Research and Analytics, National University of Singapore, Singapore [2]Department of Industrial Engineering and Decision Analytics, the Hong Kong University of Science and Technology, Hong Kong, China [3]Department of Industrial Systems Engineering and Management, National University of Singapore, Singapore. Correspondence to: Jiashuo Jiang <jsjiang@ust.hk>, Wang Chi Cheung <isecwc@nus.edu.sg>.

*Proceedings of the 43$^{rd}$ International Conference on Machine Learning*, Seoul, South Korea. PMLR 306, 2026. Copyright 2026 by the author(s).

and serves as an essential tool for stochastic control and reinforcement learning (Puterman, 2014; Bertsekas, 2012; Sutton & Barto, 2018). One of the most important problems is to learn an approximately optimal policy for an MDP with high probability, given access only to a generative model. This involves identifying a near-optimal policy in terms of expected cumulative reward over an infinite horizon through sampling state transitions. Two main classes have been extensively studied: discounted MDPs (DMDPs) (Wang, 2020; Jin & Sidford, 2020; Sidford et al., 2018; Wainwright, 2019) and average-reward MDPs (AMDPs) (Wang, 2017; Jin & Sidford, 2020; 2021).

In this paper, we focus on the specific problem of infinite-horizon DMDPs. A DMDP instance is described as a tuple $\mathcal{M} = (\mathcal{S}, \mathcal{A}, P, r, \gamma)$. Specifically, $\mathcal{S}$ is a known finite state space with size $|\mathcal{S}|$. The collection $\mathcal{A} = \{\mathcal{A}_i\}_{i\in\mathcal{S}}$ is a known collection of finite action sets $\mathcal{A}_i$ for each state $i \in \mathcal{S}$. For simplicity, we denote $\mathcal{N} = \{(i,a) : i \in \mathcal{S}, a \in \mathcal{A}_i\}$ as the set of all state-action pairs, with size $|\mathcal{N}| = N$. The matrix $P = (p(j \mid i, a))_{(i,a)\in\mathcal{N}, j\in\mathcal{S}} \in \mathbb{R}^{N\times|\mathcal{S}|}$ is the unknown state-action-state transition matrix. The $((i,a),j)$-th entry $p(j \mid i, a) \in [0, 1]$ represents the transition probability from state $i$ to state $j$ under action $a$. For any state-action pair $(i, a) \in \mathcal{N}$, we have $\sum_{j\in\mathcal{S}} p(j \mid i, a) = 1$. The array $r = (r_{i,a})_{i\in\mathcal{S}, a\in\mathcal{A}_i}$ is the known vector of state-action rewards, where $r_{i,a} \in [0, 1]$ denotes the reward received when taking action $a$ at state $i$. Finally, $\gamma \in (0, 1)$ is the known discount factor of the MDP. Following the literature, we assume that we access to a generative model, which allows us to sample state transitions from the DMDP model. In this setting, the sample complexity refers to the total number of samples required to compute a policy whose expected cumulative reward is within $\epsilon$ of optimality. The objective is to minimize this sample complexity.

This problem has been extensively studied in the literature. (Gheshlaghi Azar et al., 2013) establishes a sample complexity lower bound of $\tilde{\Omega}((1-\gamma)^{-3}N\epsilon^{-2})$ for finding an $\epsilon$-optimal policy. This bound is achieved by methods such as variance-reduced value iteration (Sidford et al., 2018) and Q-learning (Wainwright, 2019). Additionally, (Jin & Sidford, 2020) proposes a stochastic mirror descent algorithm for both DMDPs and AMDPs and achieves a sample complexity bound of $\tilde{O}((1-\gamma)^{-4}N\epsilon^{-2})$ for DMDPs, which

currently represents the best achievable sample complexity bound among primal-dual-type methods.

Motivated by the Online Algorithms with Advice framework (Mitzenmacher & Vassilvitskii, 2022) and reinforcement learning (RL) with Advice (Golowich & Moitra, 2022), we consider learning infinite-horizon DMDPs with possibly biased predictions. The process consists of two phases: a preparation phase and a learning phase. In the preparation phase, we receive a *prediction* matrix $\hat{P}$ for the latent transition matrix P, without knowledge of the prediction error. The subsequent learning phase mirrors the standard problem of learning infinite-horizon DMDPs with a generative model, with the added feature that the prediction matrix $\hat{P}$ can be incorporated into decision making. The goal remains to maximize the expected cumulative discounted reward over an infinite horizon.

Intuitively, when the prediction $\hat{P}$ is inaccurate, it should be ignored entirely. Conversely, when $\hat{P}$ is sufficiently close to P, it can be leveraged to reduce unnecessary sampling. For example, in the extreme case where $\hat{P} = P$, the optimal policy can be obtained directly through value iteration, policy iteration, or linear programming (Puterman, 2014), eliminating the need for any additional sampling. The above discussion raises the following question (Que): *Can we design an algorithm that effectively leverages the prediction to improve the sample complexity bound, yet remains robust when the prediction is missing or inaccurate—without knowing its quality?*

However, the answer is nuanced. To address this question, we make the following novel contributions:

**Impossibility Result.** In Section 3, we demonstrate that, even with predictions, no algorithm can compute an $\epsilon$-optimal policy with fewer than or equal to $\tilde{o}((1 - \gamma)^{-3} N \epsilon^{-2})$ samples for all DMDP instances without prior knowledge of the prediction error of $\hat{P}$. This result implies that no algorithm utilizing predictions can outperform the approaches proposed by (Sidford et al., 2018; Wainwright, 2019; Agarwal et al., 2020a), which achieve a sample complexity bound of $\tilde{O}((1 - \gamma)^{-3} N \epsilon^{-2})$.

**Leverage Predictions without Knowing the prediction error.** In Section 4, we take a small step back and focus on primal-dual type algorithms. We propose Optimistic-Predict Mirror Descent (OpPMD), for computing an $\epsilon$-optimal policy, while judiciously utilizing predicted transition matrix $\hat{P}$. OpPMD incorporates the prediction into the estimation of future gradients in a novel way, using carefully designed learning rates that eliminate the need for prior knowledge of the prediction error. We derive a sample complexity bound for OpPMD that depends on the prediction error, despite the algorithm not knowing this error. This bound is uniformly better than $\tilde{O}((1 - \gamma)^{-4} N \epsilon^{-2})$, the current state-of-the-

art bound achieved by primal-dual methods. Moreover, in some special cases, this bound improves upon the minimax sample complexity bound $\tilde{O}((1 - \gamma)^{-3} N \epsilon^{-2})$.

Finally, we provide numerical experiments to validate our approach in Appendix D. The result demonstrates the benefits of predictions and illustrates how effectively our algorithm leverages the prediction judiciously. We further provide discussions on future directions in Appendix A.4.

## 1.1. More about Prediction $\hat{P}$ and its applications

In this paper, we primarily investigate whether predictions of transition matrix P can enhance the process of computing an (approximately) optimal policy and improve the sample complexity bound. The prediction $\hat{P}$ is expressed as a matrix $\hat{P} = (\hat{p}(j \mid i, a))_{(i,a)\in\mathcal{N}, j\in\mathcal{S}} \in \mathbb{R}^{|\mathcal{N}|\times|\mathcal{S}|}$, with the same dimensions as the true transition matrix P. Here, $\hat{p}(j \mid i, a) \in [0, 1]$ for all $i, a, j$ and $\sum_{j\in\mathcal{S}} \hat{p}(j \mid i, a) = 1$ for all $(i, a) \in \mathcal{N}$. The predicted transition matrix $\hat{P}$ is provided during the preparation phase, prior to the standard sampling and learning phase. The prediction error $\text{Dist}(\cdot, \cdot)$ is defined as follows, and is **not known in advance**:

$$\text{Dist}(P, \hat{P}) = \max_{(i,a)\in\mathcal{N}} \sum_{j\in\mathcal{S}} |\hat{p}(j \mid i, a) - p(j \mid i, a)|.$$

The prediction is a general black-box model, and we do not impose any additional assumptions on $\hat{P}$. Besides, $\hat{P}$ is allowed to be arbitrarily biased or adversarial.

In practice, such predictions can often be obtained from external sources. There are several natural candidates for constructing a prediction $\hat{P}$. For example, suppose we already have a collection of transition observations $\{(i_\ell, a_\ell, j_\ell)\}_{\ell\in[T_0]}$ generated by an MDP whose transition matrix is possibly close to that of the DMDP model we aim to learn. In this case, we can estimate $\hat{P}$ as the empirical mean for each tuple $(i, a, j)$:

$$\hat{p}(j \mid i, a) = \frac{|\{\ell : (i_\ell, a_\ell, j_\ell) = (i, a, j)\}|}{\max\{|\{\ell : (i_\ell, a_\ell) = (i, a)\}|, 1\}}.$$

Such auxiliary datasets are widely available in real-world applications, such as simulation engines in robotics (Todorov et al., 2012; Lazaridis et al., 2020) and historical records in healthcare (Chen et al., 2022; Mate et al., 2022).

Moreover, in transfer and multi-task reinforcement learning (RL) (Zhu et al., 2023; Hua et al., 2021), knowledge acquired from related tasks can also be leveraged for future learning. In addition, the model-based RL literature provides methods for estimating transition matrices directly from data, such as those in (Agarwal et al., 2020a; Li et al., 2020). Further discussion is provided in Appendix A.1.

## 1.2. Techniques Overview on Leveraging $\hat{P}$?

Following the line of (Wang, 2017; 2020; Jin & Sidford, 2020), we formulate the infinite-horizon DMDP as a mini-max bilinear optimization problem and propose a specific primal-dual type algorithm that integrates prediction. To leverage the prediction effectively, we introduce several novel improvements. First, we construct predicted gradients based on the prediction $\hat{P}$, using it as a proxy for future gradients. Together with the standard approach of constructing stochastic unbiased gradients from sampled state transitions, our method introduces a novel combination of these two gradient sources. This approach ensures that we benefit from accurate predictions while maintaining the same convergence rate as prediction-free cases when the prediction is uninformative. Second, by tuning the learning rate, our algorithm becomes parameter-free, removing the dependence on the prediction error in implementation. Third, we provide a new variance reduction technique to control the variance of stochastic gradients estimators and shrink it as the sample size increases, ensuring efficiency. Together, we provide a robust primal-dual type algorithm, showing that it can efficiently leverage the prediction $\hat{P}$ to achieve a more favorable sample complexity bound without knowing the prediction error.

## 1.3. Related Work

Dynamic Programming and Reinforcement Learning have been well studied for decades with numerous applications in industry and finance. For further details, see textbooks (Bertsekas, 1996; 2012; 2022; Sutton & Barto, 2018; Szepesvári, 2022) and surveys (Kaelbling et al., 1996; Arulkumaran et al., 2017).

Solving MDPs with access only to a generative model is one of the most classical problems in dynamic programming and reinforcement learning. Both DMDPs and AMDPs have been well studied under this assumption. For DMDPs, (Gheshlaghi Azar et al., 2013) provides a sample complexity lower bound of $\tilde{\Omega}((1-\gamma)^{-3}N\epsilon^{-2})$. There are several works that match this bound, including value iteration (Sidford et al., 2018), Q-learning (Wainwright, 2019), and empirical MDP with a black-box model (Agarwal et al., 2020a). However, no primal-dual algorithm achieves this bound. (Wang, 2020) proposes a randomized primal-dual method with sample complexity $\tilde{\Omega}(\min\{(1-\gamma)^{-6}|\mathcal{S}|^2N\epsilon^{-2}, \tau^4(1-\gamma)^{-4}N\epsilon^{-2}\})$, where $\tau$ is an ergodic condition parameter. (Jin & Sidford, 2020; Cheng et al., 2020) achieve $\tilde{O}((1-\gamma)^{-4}N\epsilon^{-2})$ using mirror-descent-type algorithms, representing the best results for primal-dual methods. Our work is closely related to (Jin & Sidford, 2020), and we provide more discussions in Appendix A.2. Whether convex optimization methods can achieve the optimal sample complexity remains an open

problem. Our work takes a different approach: incorporating predictions of the transition matrix to improve sample complexity bounds. To the best of our knowledge, this is the first work to introduce the notion of prediction into solving MDPs with access only to a generative model.

Our work is closely related to RL with prediction. Closest to our setting, (Feng et al., 2019) assumes full access to an approximate model $\mathcal{M}_0$ with a known distance to the true model $\mathcal{M}$, bounded in terms of total variation (TV) distance. This assumption enables structure-based techniques such as suboptimal actions elimination. In contrast, our setting assumes access only to a black-box prediction of the transition matrix, with no assumptions, or knowledge of, how accurate it is. (Golowich & Moitra, 2022) studies online tabular MDPs with advice on the Q-values. Similarly, it requires knowledge of the error of the advice. (Li et al., 2024) studies single-trajectory time-varying MDPs with untrusted machine-learned predictions. (Cutkosky et al., 2022; Lyu & Cheung, 2023) leverage predictions in stochastic linear bandits and non-stationary bandits with knapsack, respectively. However, all these works focus on regret. In contrast, our work is the first to focus on utilizing black-box predictions to reduce exploration and improve sample complexity in solving MDPs. More importantly, unlike (Feng et al., 2019; Golowich & Moitra, 2022), our approach eliminates the need to know a nontrivial upper bound on the prediction error. Our work also aligns with the broader line of research on Algorithms with Advice (Mitzenmacher & Vassilvitskii, 2022) in general, and we provide more discussions in Appendix A.3.

Our algorithmic design is inspired by the parameter-free methods and adaptive learning rates from deterministic optimization. (Carmon & Hinder, 2022) develop a parameter-free stochastic gradient descent algorithm that achieves a double-logarithmic factor overhead compared to the optimal rate in the known-parameter setting. (Streeter & McMahan, 2010) introduces adaptive learning rates for online gradient descent. Interested readers can consult (Auer et al., 2002; Orabona, 2014; Cutkosky & Orabona, 2018; Orabona, 2019) for more references.

Our work is also related to the line of Offline Reinforcement Learning (Levine et al., 2020). In this setting, the learning algorithm cannot interact with the environment and collect additional information about the model. Instead, it is provided by a static dataset of transitions and must learn the best policy based on this dataset. Various approaches have been developed for this problem, such as (Fujimoto et al., 2019; Kumar et al., 2019; 2020; Agarwal et al., 2020b; Wu et al., 2019). While such datasets could potentially be used to construct a prediction transition matrix (e.g., via sample mean), these methods cannot apply to our setting, as our model allows for interaction with the environment through

sampling state transitions. Furthermore, these approaches typically leverage datasets in ways far from estimating the transition matrix, limiting their applicability to our problem, where no dataset is available in advance.

### 1.4. Notations

For a positive integer $n$, we denote $[n] = \{1, 2, \ldots, n\}$, and let $\Delta^n = \{\boldsymbol{x} \in \mathbb{R}^n : \boldsymbol{x} \geq \boldsymbol{0}, \sum_{i=1}^{n} x_i = 1\}$ denote the $n$-dimensional probability simplex. We use the notation $O(\cdot)$, $o(\cdot)$, and $\Omega(\cdot)$ as defined in (Cormen et al., 2022). We adopt the notation $\tilde{O}(\cdot)$, $\tilde{o}(\cdot)$, and $\tilde{\Omega}(\cdot)$, which have the same meaning as $O(\cdot)$, $o(\cdot)$, and $\Omega(\cdot)$, respectively, except that they hide the logarithmic terms.

## 2. Preliminaries

### 2.1. Value function

Consider a DMDP instance $\mathcal{M} = (\mathcal{S}, \mathcal{A}, \mathrm{P}, \mathrm{r}, \gamma)$. A stationary (and randomized) policy can be represented by a collection of probability distributions $\pi = \{\pi_i\}_{i \in \mathcal{S}}$, where $\pi_i \in \Delta^{|\mathcal{A}_i|}$ is a probability distribution over $\mathcal{A}_i$, and $\pi_i(a)$ denotes the probability of taking action $a \in \mathcal{A}_i$ at state $i$. For a policy $\pi$, we define the value function vector $\boldsymbol{v}^\pi = (v_i^\pi)_{i \in \mathcal{S}} \in \mathbb{R}^{|\mathcal{S}|}$ as

$$v_i^\pi = \mathbb{E}^\pi \left[ \sum_{t=0}^{\infty} \gamma^t \mathrm{r}_{i_t, a_t} | i_0 = i \right], \quad \forall i \in \mathcal{S}.$$

The expectation operator $\mathbb{E}^\pi$ is taken over the state-action trajectory $(i_0, a_0, i_1, a_1, \ldots)$ generated by the MDP under policy $\pi$. The optimal value vector $\boldsymbol{v}^* = (v_i^*)_{i \in \mathcal{S}} \in \mathbb{R}^{|\mathcal{S}|}$ is defined, for all $i \in \mathcal{S}$, as

$$v_i^* = \max_\pi \mathbb{E}^\pi \left[ \sum_{t=0}^{\infty} \gamma^t \mathrm{r}_{i_t, a_t} | i_0 = i \right]$$
$$= \mathbb{E}^{\pi^*} \left[ \sum_{t=0}^{\infty} \gamma^t \mathrm{r}_{i_t, a_t} | i_0 = i \right].$$

where $\pi^*$ is an optimal stationary policy that achieves $\boldsymbol{v}^*$. Given an initial distribution $\boldsymbol{q} \in \Delta^{|\mathcal{S}|}$, the value function of a policy $\pi$ with respect to $\boldsymbol{q}$ is defined as

$$v^\pi(\boldsymbol{q}) = \mathbb{E}^\pi \left[ \sum_{t=0}^{\infty} \gamma^t \mathrm{r}_{i_t, a_t} | i_0 \sim \boldsymbol{q} \right] = \boldsymbol{q}^\top \boldsymbol{v}^\pi.$$

with its optimal $v^*(\boldsymbol{q}) = \boldsymbol{q}^\top \boldsymbol{v}^*$. A policy $\pi^{(\epsilon)}$ is $\epsilon$-optimal with respect to $\boldsymbol{q}$ if $v^*(\boldsymbol{q}) \leq v^{\pi^{(\epsilon)}}(\boldsymbol{q}) + \epsilon$.

### 2.2. Bellman Equation, Linear Programming (LP) Formulation and Minimax Formulation

According to (Bertsekas, 2012; Puterman, 2014), $\boldsymbol{v}^*$ is the optimal value vector of a DMDP if and only if it satisfies

the following *Bellman equations*

$$v_i^* = \max_{a \in \mathcal{A}_i} \left\{ \gamma \sum_{j \in \mathcal{S}} p(j \mid i, a) v_j^* + \mathrm{r}_{i,a} \right\}, \quad \forall i \in \mathcal{S}. \quad (1)$$

When $\gamma \in (0, 1)$, the Bellman equation has a unique fixed-point solution. A policy $\pi^*$ is an optimal policy for the DMDP if it achieves $\boldsymbol{v}^*$ coordinate-wise, i.e. $\boldsymbol{v}^* = \boldsymbol{v}^{\pi^*}$. For finite-state DMDPs, such an optimal policy $\pi^*$ always exists. (Puterman, 2014) shows that the Bellman equation (1) is equivalent to the following linear programming (LP) problem:

$$\min_{\boldsymbol{v} \in \mathbb{R}^{|\mathcal{S}|}} \quad (1 - \gamma)\boldsymbol{q}^\top \boldsymbol{v}, \quad \text{s.t.} \quad (\hat{\mathrm{I}} - \gamma \mathrm{P})\boldsymbol{v} - \mathrm{r} \geq \boldsymbol{0}. \quad (2)$$

where $\hat{\mathrm{I}} \in \mathbb{R}^{|\mathcal{N}| \times |\mathcal{S}|}$ is the matrix defined as $I_{(i,a),j} = \mathbf{1}[i = j]$ for each $((i, a), j) \in \mathcal{N} \times \mathcal{S}$. The dual problem of (2) is

$$\max_{\boldsymbol{\mu} \in \Delta^N} \quad \boldsymbol{\mu}^\top \mathrm{r}, \quad \text{s.t.} \quad (\hat{\mathrm{I}} - \gamma \mathrm{P})^\top \boldsymbol{\mu} = (1 - \gamma)\boldsymbol{q}. \quad (3)$$

Let $\boldsymbol{\mu}^*$ denote an optimal solution of (3). We can formulate (2), (3) as the following equivalent minimax problem

$$\min_{\boldsymbol{v} \in \mathcal{V}} \max_{\boldsymbol{\mu} \in \mathcal{U}} \quad f(\boldsymbol{v}, \boldsymbol{\mu}),$$
$$\text{s.t.} \quad f(\boldsymbol{v}, \boldsymbol{\mu}) = (1 - \gamma)\boldsymbol{q}^\top \boldsymbol{v} + \boldsymbol{\mu}^\top ((\gamma \mathrm{P} - \hat{\mathrm{I}})\boldsymbol{v} + \mathrm{r}). \quad (4)$$

where $\mathcal{V} = \{\boldsymbol{v} \in \mathbb{R}^{|\mathcal{S}|} : \|\boldsymbol{v}\|_\infty \leq (1 - \gamma)^{-1}\}, \mathcal{U} = \Delta^N$. It is straightforward to verify that $\boldsymbol{v}^* \in \mathcal{V}$ and $\boldsymbol{\mu}^* \in \mathcal{U}$. Since $\mathrm{r}_{i,a} \in [0, 1]$ for all $(i, a)$, it follows that $v_i^* \leq (1 - \gamma)^{-1}$ for all $i \in \mathcal{S}$. Moreover, multiplying $\boldsymbol{e}^\top$ (where $\boldsymbol{e} = (1, \ldots, 1)$) on both sides of the constraint $(\hat{\mathrm{I}} - \gamma \mathrm{P})^\top \boldsymbol{\mu} = (1 - \gamma)\boldsymbol{q}$, we can confirm that $\boldsymbol{e}^\top \boldsymbol{\mu}^* = 1$, as $\boldsymbol{q}$ is a probability distribution.

## 3. An Impossibility Result

We demonstrate that even with predictions, no algorithm can compute an $\epsilon$-optimal policy for all DMDP instances using strictly fewer than $\tilde{O}((1 - \gamma)^{-3} N \epsilon^{-2})$ samples without knowledge of the prediction error, i.e., the discrepancy between $\hat{\mathrm{P}}$ and the true transition matrix P. The analysis is inspired by the lower bound in the prediction-free setting in (Gheshlaghi Azar et al., 2013), as well as high-level ideas from the Pareto regret frontier for multi-armed bandits (Lattimore, 2015). We focus on $(\epsilon, \delta)$-*smart* algorithm whose objective is to output a near-optimal policy with high probability through sampling, as defined below:

**Definition 3.1.** We say that an algorithm ALG is an $(\epsilon, \delta)$-*smart* algorithm with respect to $T$ on a DMDP model $\mathcal{M}$ if it outputs a value function vector $\hat{\boldsymbol{v}}_T$ such that

$$\Pr_{\mathcal{M}, T} (\|\hat{\boldsymbol{v}}_T - \boldsymbol{v}^*\|_\infty > \epsilon) < \delta. \quad (5)$$

Here, $\hat{v}_T$ is generated by ALG after adaptively sampling $T$ state-action pairs and observing the subsequent transitions from the model $\mathcal{M}$.

An $(\epsilon, \delta)$-*smart* algorithm guarantees finding an $\epsilon$-optimal value function with high probability without knowing the true model $\mathcal{M}$, rather than an $\epsilon$-optimal policy with high probability. Notably, while an $\epsilon$-optimal value function does not necessarily imply an $\epsilon$-optimal policy, the converse is true. Therefore, a lower bound on the sample complexity for finding an $\epsilon$-optimal value function also serves as a lower bound for finding an $\epsilon$-optimal policy. We are now ready to state our impossibility result.

**Theorem 3.2.** *Suppose $N \geq 6$, $\gamma \in [1/3, 1)$, $\epsilon \in \left(0, (1-\gamma)^{-1}/40\right]$, and $\delta \in (0, 0.24]$. Consider a fixed but arbitrary algorithm ALG. If ALG is $(\epsilon, \delta)$-smart on a specific DMDP instance $\mathcal{M}_0$ with state space $\mathcal{S}$, action apace $\mathcal{A}$, a total $N$ state-action pairs, discount factor $\gamma$, and transition matrix $P_0$ given access to a black-box prediction $\hat{P}$ satisfying $\hat{P} = P_0$, and the number of samples $T$ satisfies*

$$T \leq \frac{1}{300}(1-\gamma)^{-3}\left(\frac{N}{3} - 1\right)\epsilon^{-2}\ln\left(\frac{1}{4.1\delta}\right), \quad (6)$$

*then there exists another DMDP instance $\mathcal{M}'$ with the same $\mathcal{S}$, $\mathcal{A}$, and $\gamma$ as $\mathcal{M}_0$ such that, even with the same prediction $\hat{P}$, ALG fails to be $(\epsilon, \delta)-$smart on $\mathcal{M}'$ with the same sample size $T$.*

Theorem 3.2 is proved in Appendix B.2. In fact, even if ALG knows that the instance is either $\mathcal{M}_0$ or $\mathcal{M}'$, which share the same prediction $\hat{P}$, but does not know the exact identity of the instance, the impossibility result still holds. This highlights the fundamental challenge of leveraging predictions.

While our lower bound proof shares some similarities with (Gheshlaghi Azar et al., 2013), the core problem we address is fundamentally different. (Gheshlaghi Azar et al., 2013) establishes a sample complexity lower bound for estimating an $\epsilon$-optimal value function in DMDPs with a generative model, whereas we introduce a prediction matrix representing prior knowledge about the model with unknown quality. In particular, we ask whether it is possible for an algorithm to be simultaneously

(a) $(\epsilon, \delta)$-smart on instance $\mathcal{M}_0$ with $T = T_{smart}$ for some $T_{smart} = \tilde{o}((1-\gamma)^{-3}N\epsilon^{-2})$,

(b) $(\epsilon, \delta)$-smart with $T = \tilde{O}((1-\gamma)^{-3}N\epsilon^{-2})$ on all instances, including the hard instance $\mathcal{M}'$.

By (Gheshlaghi Azar et al., 2013), achieving (b) is possible, and $\tilde{O}((1-\gamma)^{-3}N\epsilon^{-2})$ characterizes the minimax sample

complexity. However, we focus on the joint feasibility of (a) and (b), and in particular on how small $T_{smart}$ can be in (a) while still guaranteeing (b). This consideration involves a modeling assumption different from existing minimax sample complexity bounds (Gheshlaghi Azar et al., 2013), since in our setting the learner is equipped with a prediction $\hat{P}$. Accordingly, we must carefully analyze how $\hat{P}$ interacts with the sample complexity lower bound.

The impossibility of achieving (a) and (b) simultaneously is not obvious. A natural approach would be to test whether the latent P is "close" to $\hat{P}$ using $T_{smart}$ samples, with the hope that $T_{smart} = o((1-\gamma)^{-3}N\epsilon^{-2})$. If so, one can adopt (a) by directly using $\hat{P}$ to construct a value function. Otherwise, one can resort to a minimax algorithm, such as those in (Sidford et al., 2018; Wainwright, 2019) to achieve (b). The crucial question is therefore whether a sufficiently powerful test exists to achieve $T_{smart} = o((1-\gamma)^{-3}N\epsilon^{-2})$. Theorem 3.2 formally rules out this possibility, and establishes a connection between the possibility of such tests and the change-of-measure argument between instances.

Finally we remark that the condition $\gamma \geq 1/3$ is not a fundamental barrier. To obtain a meaningful lower bound, it suffices to require $\gamma \geq c$ for some constant $c$ that $c > 0$. In the current Theorem 3.2, we choose $c = 1/3$ for convenience. If a smaller $c$ is chosen, the result still holds with a smaller absolute constant (smaller than current $1/300$) in the bound. This condition is merely a technical convenience rather than a fundamental restriction.

## 4. Leverage Black-Box Prediction $\hat{P}$

We propose an algorithm, Optimistic-Predict Mirror Descent (OpPMD), presented in Algorithm 1, that leverages the prediction $\hat{P}$ without requiring knowledge of the prediction error. Our approach hinges on a carefully designed primal-dual mirror descent method for solving the minimax problem (4) while incorporating $\hat{P}$. In Section 4.1, we provide the gradient estimators required for mirror descent on $v$- and $\mu$-sides, respectively. Section 4.2 details OpPMD. In Section 4.3, we analyze the algorithm, providing a sample complexity bound that depends on the prediction error $\text{Dist}(P, \hat{P})$, even though OpPMD does not know it. For simplicity, throughout this section we abbreviate $\text{Dist}(P, \hat{P})$ as Dist.

### 4.1. Gradient Estimators

The minimax optimization procedure requires access to the gradients with respect to $v$ and the negative gradients with respect to $\mu$ of the function $f(\cdot, \cdot)$. Specifically, for each pair $(v, \mu)$, the following quantities are required: For the

**Algorithm 1** Optimistic-Predict-Mirror Descent (OpPMD)

1: **Input:** Optimization length $T$, initialized $\boldsymbol{v}_1 \in \mathcal{V}$, $\boldsymbol{\mu}_1 = (\frac{1}{N}, \cdots \frac{1}{N}) \in \mathcal{U}$, initial distribution $\boldsymbol{q} \in \Delta^{|\mathcal{S}|}$, prediction matrix $\hat{\mathrm{P}}$, $\bar{\boldsymbol{g}}_1^{\boldsymbol{\mu}} = (\hat{\mathrm{I}} - \gamma\hat{\mathrm{P}})\boldsymbol{v}_1 - \mathrm{r}$.

2: **for** $t \in [T]$ **do**

3:  Sample and compute $\tilde{\boldsymbol{g}}_t^{\boldsymbol{v}}$ as (11).

4:  Compute learning rate for $\boldsymbol{v}-$side:

$$\eta_t^v = \frac{\sqrt{2}}{2} \cdot \frac{\sqrt{|\mathcal{S}|} \cdot (1-\gamma)^{-1}}{\sqrt{\sum_{i=1}^t \|\tilde{\boldsymbol{g}}_i^{\boldsymbol{v}}\|_2^2}}. \qquad (7)$$

5:  Update $\boldsymbol{v}_{t+1}$:

$$\boldsymbol{v}_{t+1} = \Pi_{\mathcal{V}}(\boldsymbol{v_t} - \eta_t^v \tilde{\boldsymbol{g}}_t^{\boldsymbol{v}}). \qquad (8)$$

6:  Sample and compute $\tilde{\boldsymbol{g}}_t^{\boldsymbol{\mu}}$ as (12).

7:  Compute predicted gradient $\bar{\boldsymbol{g}}_{t+1}^{\boldsymbol{\mu}}$ for $\boldsymbol{\mu}$ as (13).

8:  Compute learning rate for $\boldsymbol{\mu}-$side:

$$\eta_t^{\mu} = \frac{\sqrt{2}}{2} \cdot \frac{\sqrt{\ln(N)}}{\sqrt{\sum_{i=1}^t \|\tilde{\boldsymbol{g}}_i^{\boldsymbol{\mu}} - \bar{\boldsymbol{g}}_i^{\boldsymbol{\mu}}\|_\infty^2}}. \qquad (9)$$

9:  Update $\boldsymbol{\mu}_{t+1}$: $\forall \ell \in [N]$, $\mu_{t+1,\ell} =$

$$\frac{\mu_{t,\ell} \exp(-\eta_t^{\mu}(\tilde{g}_{t,\ell}^{\mu} - \bar{g}_{t,\ell}^{\mu} + \bar{g}_{t+1,\ell}^{\mu}))}{\sum_{\ell'=1}^N \mu_{t,\ell'} \exp(-\eta_t^{\mu}(\tilde{g}_{t,\ell'}^{\mu} - \bar{g}_{t,\ell'}^{\mu} + \bar{g}_{t+1,\ell'}^{\mu}))}. \qquad (10)$$

10: **end for**

11: Compute $\bar{\boldsymbol{v}} = \frac{1}{T} \sum_{t \in [T]} \boldsymbol{v}_t$, $\bar{\boldsymbol{\mu}} = \frac{1}{T} \sum_{t \in [T]} \boldsymbol{\mu}_t$.

12: Compute $\bar{\pi}$ such that

$$\bar{\pi}_{(i,a)} = \frac{\bar{\mu}_{i,a}}{\sum_{a' \in \mathcal{A}_i} \bar{\mu}_{i,a'}}.$$

13: Output $\bar{\pi}$.

---

$\boldsymbol{v}-$side and the $\boldsymbol{\mu}-$side, respectively,

$$\boldsymbol{g^v}(\boldsymbol{v}, \boldsymbol{\mu}) = \nabla_{\boldsymbol{v}} f(\boldsymbol{v}, \boldsymbol{\mu}) = (1-\gamma)\boldsymbol{q} + \boldsymbol{\mu}^\top(\gamma\mathrm{P} - \hat{\mathrm{I}}),$$

$$\boldsymbol{g^\mu}(\boldsymbol{v}, \boldsymbol{\mu}) = -\nabla_{\boldsymbol{\mu}} f(\boldsymbol{v}, \boldsymbol{\mu}) = (\hat{\mathrm{I}} - \gamma\mathrm{P})\boldsymbol{v} - \mathrm{r}.$$

However, these gradients cannot be directly computed because P is unknown. Instead, we construct several estimators that serve as proxies for the true gradients in the algorithm.

### 4.1.1. STOCHASTIC ESTIMATORS

We construct *stochastic* estimators $\tilde{\boldsymbol{g}}_t^{\boldsymbol{v}}, \tilde{\boldsymbol{g}}_t^{\boldsymbol{\mu}}$ for the gradients with respect to $\boldsymbol{v}$ and $\boldsymbol{\mu}$ via sampling. For the $\boldsymbol{v}$-side, we construct the following stochastic gradient:

Sample $(i, a) \sim \mu_{t,(i,a)}, j \sim p(j \mid i, a), i' \sim q_i$.
Compute $\tilde{\boldsymbol{g}}_t^{\boldsymbol{v}} = \tilde{\boldsymbol{g}}^{\boldsymbol{v}}(\boldsymbol{v}_t, \boldsymbol{\mu}_t) = (1-\gamma)\boldsymbol{e}_{i'} + \gamma\boldsymbol{e}_j - \boldsymbol{e}_i.$

$$(11)$$

This stochastic estimator is unbiased as shown in the following lemma.

**Lemma 4.1.** *(Lemma 3 in (Jin & Sidford, 2020)) The stochastic gradient $\tilde{\boldsymbol{g}}_t^{\boldsymbol{v}}$ for $\boldsymbol{v}$ satisfies* $\mathbb{E}[\tilde{\boldsymbol{g}}_t^{\boldsymbol{v}}] = (1-\gamma)\boldsymbol{q} + \boldsymbol{\mu}_t^\top(\gamma P - \hat{I}) = \nabla_{\boldsymbol{v}} f(\boldsymbol{v}_t, \boldsymbol{\mu}_t).$

For the $\boldsymbol{\mu}$-side, we construct the following stochastic gradient:

Sample $(i, a)$ uniformly from $\mathcal{N}$, and sample $j \sim p(j \mid i, a)$. Collect $z = (i, a, j) \in \mathcal{Z}$. Compute

$$\tilde{\boldsymbol{g}}_t^{\boldsymbol{\mu}} = \tilde{\boldsymbol{g}}^{\boldsymbol{\mu}}(\boldsymbol{v}_t, \boldsymbol{\mu}_t) = \frac{1}{t} \sum_{z=(i,a,j) \in \mathcal{Z}} N(v_{t,i} - \gamma v_{t,j} - \mathrm{r}_{i,a})\boldsymbol{e}_{i,a}.$$

$$(12)$$

This stochastic estimator is also unbiased. More importantly, its variance decreases over time by reusing past samples and averaging, ensuring greater stability and better control of the error in solving the minimax program.

**Lemma 4.2.** *The stochastic gradient $\tilde{\boldsymbol{g}}_t^{\boldsymbol{\mu}}$ for $\boldsymbol{v}$ satisfies* $\mathbb{E}[\tilde{\boldsymbol{g}}_t^{\boldsymbol{\mu}}] = (\hat{I} - \gamma P)\boldsymbol{v}_t - r = -\nabla_{\boldsymbol{\mu}} f(\boldsymbol{v}_t, \boldsymbol{\mu}_t)$ *and* $\mathbb{E}[\|\tilde{\boldsymbol{g}}_t^{\boldsymbol{\mu}} + \nabla_{\boldsymbol{\mu}} f(\boldsymbol{v}_t, \boldsymbol{\mu}_t)\|_\infty^2] \le 9N^2(1-\gamma)^{-2}/t.$

Lemma 4.1, 4.2 are proved in Appendix C.1. The proof for Lemma 4.2 is inspired by Lemma 4 in (Jin & Sidford, 2020).

### 4.1.2. PREDICTED ESTIMATORS

Motivated by the idea of optimistic mirror descent in optimization and online learning (Rakhlin & Sridharan, 2013a; Joulani et al., 2017; Orabona, 2019), we construct the following *predicted* gradient as a proxy for the next-step gradient to facilitate updating $\boldsymbol{\mu}_t$ in (10). The predicted gradient is defined as

$$\bar{\boldsymbol{g}}_{t+1}^{\boldsymbol{\mu}} = (\hat{I} - \gamma\hat{P})\boldsymbol{v}_{t+1} - \mathrm{r}. \qquad (13)$$

This predicted gradient provides additional information to the algorithm, and its usefulness depends on the accuracy of $\hat{P}$. Ideally, the true next-step gradient is $\boldsymbol{g}^{\boldsymbol{\mu}}(\boldsymbol{v}_{t+1}, \boldsymbol{\mu}_{t+1}) = (\hat{I} - \gamma P)\boldsymbol{v}_{t+1} - \mathrm{r}$. The performance of optimistic mirror descent improves significantly when $\hat{P}$ is sufficiently close to P.

### 4.2. Algorithm Description

Our algorithm is presented in Algorithm 1. Algorithm 1 consists of two parts: Line 2 to Line 11 represents the procedure for solving the minimax problem (4), and Line 12 corresponds to the process of translating a feasible and approximate solution of (4) into a policy.

In order to leverage the potential benefits of the prediction, we develop an algorithmic framework to solve the minimax problem (4) iteratively. The optimization procedure consists of $T$ steps. At step $t$, we compute the stochastic gradient

estimators $\tilde{g}_t^v$ in Line 3 for the $v$−side and $\tilde{g}_t^\mu$ in Line 6 for the $\mu$−side via sampling. For the $v$-side, we update $v_t$ according to (8). The learning rate $\eta_t^v$ defined in (7) is carefully designed such that it is agnostic to the length of the learning horizon $T$ and the desired accuracy level $\epsilon$. For the $\mu$-side, we additionally construct a *predicted* gradient in (13) using the predicted transition matrix $\hat{P}$. Then we update $\mu_t$ as in (10). Similarly, the learning rate $\eta_t^\mu$ defined in (9) is designed to be agnostic not only to $T$ and $\epsilon$, but also to the prediction error Dist.

Our approach novelly integrates two gradient sources for updating $\mu$: one derived from sampling and the other derived from predictions. Prediction-based gradients act as optimistic estimators, guiding the algorithm in regions where the predictions are accurate and thereby reducing exploration, while sample-based gradients provide unbiased updates, ensuring robustness. Notably, our algorithm is parameter-free, requiring no prior knowledge of the desired accuracy level $\epsilon$ or the prediction error Dist. This yields greater practicality in implementation and applications.

Finally, in Line 12, we adapt the approach of (Jin & Sidford, 2020) to convert an approximately optimal minimax solution into an approximately optimal policy.

### 4.3. Analysis for Sample Complexity

Before proceeding, we introduce a performance metric to evaluate the quality of any feasible solution to the minimax problem (4):

**Definition 4.3.** For the minimax problem (4), we define its *duality gap* at a given feasible pair $(v, \mu)$ as

$$\text{GAP}(v, \mu) = \max_{\mu' \in \mathcal{U}} f(v, \mu') - \min_{v' \in \mathcal{V}} f(v', \mu).$$

Moreover, an $\epsilon$-optimal solution of the minimax problem (4) is a feasible pair $(v^{(\epsilon)}, \mu^{(\epsilon)})$ such that $\text{GAP}(v^{(\epsilon)}, \mu^{(\epsilon)}) \le \epsilon$.

We are now ready to present the accuracy of the output $(\bar{v}, \bar{\mu})$ in Line 11 in terms of the duality gap.

**Lemma 4.4.** *Given the minimax problem (4), the solution $(\bar{v}, \bar{\mu})$ returned by OpPMD satisfies*

$$\mathbb{E}[GAP(\bar{v}, \bar{\mu})] \le Err_v + Err_{\mu,1} + Err_{\mu,2},$$

*where*

$$Err_v = 3(1-\gamma)^{-1}\sqrt{\frac{|\mathcal{S}|}{T}},$$

$$Err_{\mu,1} = 3\gamma(1-\gamma)^{-1}\min\{1, Dist\}\cdot\sqrt{\frac{N}{T}},$$

$$Err_{\mu,2} = 9\sqrt{2}\frac{(1-\gamma)^{-1}N\ln(T)}{T}.$$

*The expectation is taken over the randomness of the sampling procedure in OpPMD.*

Lemma 4.4 is proved in Appendix C.2. In particular, $\text{Err}_v$ arises from the error in updating $v$. $\text{Err}_{\mu,1}$ comes from the error in updating $\mu$. $\text{Err}_{\mu,1}$ can be potentially improved when the prediction error Dist is small. In other words, a possibly accurate prediction $\hat{P}$ can accelerate convergence through the mechanism of optimistic mirror descent. $\text{Err}_{\mu,2}$ is due to the variance of the stochastic gradient for $\mu$.

A key point in Lemma 4.4 is how the prediction error Dist is linked to the output solution $(\bar{v}, \bar{\mu})$, and consequently to the final policy. In particular, the carefully designed learning rate $\eta_t^\mu$ in (9), together with the weighted update in (13), ensures that the impact of the prediction error is smoothly translated to the duality gap of $(\bar{v}, \bar{\mu})$, without requiring any prior knowledge of Dist. The careful choice of update rules is a crucial step that yields an output solution $(\bar{v}, \bar{\mu})$ satisfying the desired performance guarantee, and thereby leads to a desired final policy.

We define a feasible pair $(v^{(\epsilon)}, \mu^{(\epsilon)})$ to be an *expected $\epsilon$-optimal solution* of (4) if $\mathbb{E}[\text{GAP}(v^{(\epsilon)}, \mu^{(\epsilon)})] \le \epsilon$. A policy $\pi_i^{(\epsilon)}$ is an *expected $\epsilon$-optimal policy* with respect to $q$ if $v^*(q) \le \mathbb{E}[v^{\pi^{(\epsilon)}}(q)] + \epsilon$. The expectations are taken over the randomness in the generation process of $(v^{(\epsilon)}, \mu^{(\epsilon)})$ and $\pi_i^{(\epsilon)}$. By translating an expected approximately optimal solution into an expected approximately optimal policy, we derive the following sample complexity bound, which is proved in Appendix C.3.

**Theorem 4.5.** *Given a DMDP model $\mathcal{M}$ and $\epsilon \in (0, 1)$, OpPMD in Algorithm 1 can construct an expected $\epsilon$-optimal policy $\pi^{(\epsilon)}$ using the predicted transition matrix $\hat{P}$ with sample complexity $\tilde{O}\left(\max\{T_v, T_{\mu,1}, T_{\mu,2}\}\right)$, where*

$$T_v = (1-\gamma)^{-4}|\mathcal{S}|\epsilon^{-2},$$

$$T_{\mu,1} = (1-\gamma)^{-4}N\min\{1, Dist^2\}\epsilon^{-2},$$

$$T_{\mu,2} = (1-\gamma)^{-2}N\epsilon^{-1}.$$

We make several observations regarding the sample complexity in Theorem 4.5. First, this bound improves upon the best known bound achieved by primal-dual methods $\tilde{O}((1-\gamma)^{-4}N\epsilon^{-2})$ ((Jin & Sidford, 2020)). Since our method is also a primal-dual method, this improvement highlights the value of leveraging predictions on the transition matrix, even when the prediction error is unknown.

More specifically, when the prediction is sufficiently accurate (e.g., Dist $= o(1)$), our algorithm achieves a sample complexity of $\tilde{o}((1-\gamma)^{-4}N\epsilon^{-2})$. Otherwise, it gracefully defaults to the standard performance. This demonstrates the robustness of our algorithm and its ability to safely leverage prior information without knowing its quality.

In particular, in the case where $\text{Dist}^2 \le O((1-\gamma)^2\epsilon)$ and $|\mathcal{A}| > \Omega((1-\gamma)^{-1})$, our sample complexity bound becomes

$$\tilde{O}(\max\{(1-\gamma)^{-4}|\mathcal{S}|\epsilon^{-2}, (1-\gamma)^{-2}N\epsilon^{-1}\})$$

$$= \tilde{o}((1-\gamma)^{-3}N\epsilon^{-2}), \qquad (14)$$

which is strictly better than the standard minimax bound $\tilde{O}((1-\gamma)^{-3}N\epsilon^{-2})$. This demonstrates the potential benefit of our approach when the prediction is accurate. Importantly, this does not contradict our lower bound in Theorem 3.2, as they operate in different regimes. The upper bound (14) applies only when the prediction is accurate. In this case, the algorithm does not need to distinguish between instances. In contrast, the lower bound in Theorem 3.2 reflects the worst-case performance, requiring that the algorithm must perform uniformly across all instances.

Finally, on a broader level, Theorem 4.5 illustrates how a proper and adaptive tuning of learning rates enables us to bypass the need for a non-trivial upper bound on Dist, an assumption that is required in related works such as (Golowich & Moitra, 2022; Feng et al., 2019). This connection, together with the ability to overcome the limitations of existing works, constitutes a novel aspect of our approach. It is carefully realized through the joint design of learning rates and iterates.

### 4.4. Sketch Proof for Lemma 4.4

In this subsection, we provide a sketch proof for Lemma 4.4. It is equivalent to prove a bound for $\mathbb{E}[f(\bar{\boldsymbol{v}}, \boldsymbol{\mu}) - f(\boldsymbol{v}, \bar{\boldsymbol{\mu}})]$ for any $\boldsymbol{v} \in \mathcal{V}, \boldsymbol{\mu} \in \mathcal{U}$, since

$$\mathbb{E}[\text{GAP}(\bar{\boldsymbol{v}}, \bar{\boldsymbol{\mu}})] = \mathbb{E}[\max_{\boldsymbol{\mu}'} f(\bar{\boldsymbol{v}}, \boldsymbol{\mu}') - \min_{\boldsymbol{v}'} f(\boldsymbol{v}', \bar{\boldsymbol{\mu}})]$$
$$= \mathbb{E}[f(\bar{\boldsymbol{v}}, \bar{\boldsymbol{\mu}}') - f(\bar{\boldsymbol{v}}', \bar{\boldsymbol{\mu}})],$$

$\bar{\boldsymbol{\mu}}' \in \arg\max_{\boldsymbol{\mu} \in \mathcal{U}} f(\bar{\boldsymbol{v}}, \boldsymbol{\mu}), \bar{\boldsymbol{v}}' \in \arg\max_{\boldsymbol{v} \in \mathcal{V}} f(\boldsymbol{v}, \bar{\boldsymbol{\mu}})$. We can decompose it as follows: $\mathbb{E}[f(\bar{\boldsymbol{v}}, \boldsymbol{\mu}) - f(\boldsymbol{v}, \bar{\boldsymbol{\mu}})] =$

$$\frac{1}{T}\mathbb{E}\left[\sum_{t=1}^{T} \tilde{\boldsymbol{g}}^{\boldsymbol{\mu}}(\boldsymbol{v}_t, \boldsymbol{\mu}_t)^\top \boldsymbol{\mu}_t - \sum_{t=1}^{T} \tilde{\boldsymbol{g}}^{\boldsymbol{\mu}}(\boldsymbol{v}_t, \boldsymbol{\mu}_t)^\top \boldsymbol{\mu}\right] \quad (15a)$$

$$+\frac{1}{T}\mathbb{E}\left[\sum_{t=1}^{T} \tilde{\boldsymbol{g}}^{\boldsymbol{v}}(\boldsymbol{v}_t, \boldsymbol{\mu}_t)^\top \boldsymbol{v}_t - \sum_{t=1}^{T} \tilde{\boldsymbol{g}}^{\boldsymbol{v}}(\boldsymbol{v}_t, \boldsymbol{\mu}_t)^\top \boldsymbol{v}\right]. \quad (15b)$$

(15a) reflects the error from updating $\boldsymbol{\mu}$ and (15b) reflects the error from updating $\boldsymbol{v}$. We analyze these two terms respectively. For (15a), $T \cdot (15a)$

$$= \mathbb{E}\left[\frac{\ln(N)}{\eta_T^{\boldsymbol{\mu}}} + \frac{1}{2}\sum_{t=1}^{T} \eta_t^{\boldsymbol{\mu}}\|\tilde{\boldsymbol{g}}_t^{\boldsymbol{\mu}} - \bar{\boldsymbol{g}}_t^{\boldsymbol{\mu}}\|_\infty^2\right] \quad (16a)$$

$$= O\left(\sqrt{\ln(N)}\right) \cdot \left(\mathbb{E}\left[\sqrt{\sum_{t=1}^{T} \|\tilde{\boldsymbol{g}}_t^{\boldsymbol{\mu}} - \bar{\boldsymbol{g}}_t^{\boldsymbol{\mu}}\|_\infty^2}\right]\right.$$

$$\left.+\mathbb{E}\left[\sum_{t=1}^{T} \frac{\|\tilde{\boldsymbol{g}}_t^{\boldsymbol{\mu}} - \bar{\boldsymbol{g}}_t^{\boldsymbol{\mu}}\|_\infty^2}{\sqrt{\sum_{i=1}^{t}\|\tilde{\boldsymbol{g}}_i^{\boldsymbol{\mu}} - \bar{\boldsymbol{g}}_i^{\boldsymbol{\mu}}\|_\infty^2}}\right]\right) \quad (16b)$$

$$\leq O\left(\sqrt{\ln(N)}\right) \cdot \mathbb{E}\left[\sqrt{\sum_{t=1}^{T} \|\tilde{\boldsymbol{g}}_t^{\boldsymbol{\mu}} - \bar{\boldsymbol{g}}_t^{\boldsymbol{\mu}}\|_\infty^2}\right] \quad (16c)$$

$$\leq O\left(\sqrt{\ln(N)}\right) \cdot \left(\sqrt{\mathbb{E}\left[\sum_{t=1}^{T} \|\tilde{\boldsymbol{g}}_t^{\boldsymbol{\mu}} - \boldsymbol{g}^{\boldsymbol{\mu}}(\boldsymbol{v}_t, \boldsymbol{\mu}_t)\|_\infty^2\right]}\right.$$

$$\left.+\sqrt{\mathbb{E}\left[\sum_{t=1}^{T} \|\boldsymbol{g}^{\boldsymbol{\mu}}(\boldsymbol{v}_t, \boldsymbol{\mu}_t) - \bar{\boldsymbol{g}}_t^{\boldsymbol{\mu}}\|_\infty^2\right]}\right) \quad (16d)$$

$$\leq T \cdot \tilde{O}\left(\text{Err}_{\mu,2} + \text{Err}_{\mu,1}\right).$$

(16a) comes from the performance guarantee for optimistic (online) mirror descent. (16b) comes from the definition of learning rate $\eta_t^{\boldsymbol{\mu}}$, (16c) comes from Lemma 4.13 in (Orabona, 2019). (16d) comes from Jensen inequality and triangle inequality. We remark that (16b) and (16c) demonstrates how the adaptive learning rates work. Follow the similar line, we can derive that

$$T \cdot (15b) \leq T \cdot O\left(\text{Err}_v\right).$$

Altogether, the Theorem is proved.

## 5. Conclusion

In this paper, we study how to safely and effectively leverage black-box predictions of the transition matrix without knowing their accuracy. Building on primal-dual methods, our approach adaptively incorporates the prediction and improves upon the best known bounds for such methods. When the prediction is accurate, our algorithm achieves a better sample complexity than the minimax bound; otherwise, it gracefully defaults to standard performance, all without requiring knowledge of the prediction quality.

There are several promising future directions. While our work focuses on the tabular setting, an interesting next step is to extend our framework to more complex RL environments. Specifically, the gradient-based structure of our algorithm makes it compatible with function approximation techniques, which are essential for linear MDP and other large-scale applications. We provide further discussions in Appendix A.4.

## Acknowledgements

Wang Chi would like to acknowledge the support from the Singapore Ministry of Education AcRF Tier 2 Grant (Proposal ID: T2EP20124-0037, Award number MOE-T2EP20124-0008). This research work is partially funded by Institute of Operations Research and Analytics, National University of Singapore. Lixing would like to acknowledge the support from NUS Research Scholarship. Jiashuo would like to acknowledge the support from the Early Career Scheme (ECS) 26210223 and the General Research Fund (GRF) 16204024 from the Research Grants Council, Hong Kong.

## Impact Statement

This paper presents work whose goal is to advance the field of Reinforcement Learning and Optimization. We primarily focus on theoretical research. There are many potential societal consequences of our work, none which we feel must be specifically highlighted here.

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

# A. More Discussions on Existing Works and Future Directions

## A.1. More Discussions on How to Construct Prediction Transition Matrix $\hat{P}$

In many applications of MDPs and Reinforcement Learning, the prediction transition matrix $\hat{P}$ can be readily available or can be directly constructed. The applications has been previously explored by (Golowich & Moitra, 2022), which studies online tabular finite-horizon episodic MDPs with predictions in Q-value. Our framework expands the research landscape by (1) incorporating predictions to solve infinitie-horizon MDPs and improve sample complexity bounds, whereas prior work has primarily focused on regret, and (2) exploring predictions on the transition matrix rather than on the Q-value, which is often easier to obtain in practice. In the following, we explain the rationale behind our model. Interesting readers can consult (Golowich & Moitra, 2022) for more discussions.

**Physics Simulation** There are several powerful physics simulation engines in robotics, such as MuJoCo (Todorov et al., 2012; Lazaridis et al., 2020; Tessler et al., 2019; Weng et al., 2022), which allow us to collect huge amount of data from simulations. While such data may not be directly applicable to the decision-making process due to the dynamic changing environment, it can be used to construct various predictions, such as $\hat{P}$ in our paper or Q-value in (Golowich & Moitra, 2022) to enhance learning algorithms. Several related works include (Zhao et al., 2020; Peng et al., 2018; Collins et al., 2021).

**Healthcare** In healthcare applications (Yu et al., 2021), extensive records from previous clinical trials capture treatments and outcomes from numerous patients. Although individuals differ, it is reasonable to assume that learning tasks exhibit similarities across patients. This allows for leveraging these records to construct predictions, such as transition matrices. There are some related work, such as (Chen et al., 2022; Mate et al., 2022). Additionally, various widely used health apps, such as Apple Health, MyFitnessPal, Sleep Cycle, collect vast amounts of user data that can contribute to predictive modeling and personalized insights.

**Transfer Learning and Multi-task Learning** Our framework for leveraging predictions to enhance the solving of DMDPs is related to transfer learning or multi-task learning. In the context of reinforcement learning (RL), these approaches aim to use knowledge from one RL task to improve learning efficiency and performance across several different but related RL tasks. This can significantly reduce the time, data or computational resources required to learn new tasks. The prediction matrix $\hat{P}$ in our framework can be viewed as an estimation of the transition matrix from an earlier task, which can be utilized to accelerate learning future related tasks. Interesting readers can refer to (Taylor & Stone, 2009; Zhu et al., 2023; Hua et al., 2021; Gamrian & Goldberg, 2019) for further details.

**Model-based RL** The model-based reinforcement learning approaches also provide methods for estimating transition matrices directly from data. For exmaple, (Agarwal et al., 2020a) shows how the natural plug-in MLE approach can achieve convergence and minimax sample complexity. (Li et al., 2020) proposes a perturbed model-based planning algorithm with refined analysis to remove the sample-size barrier and achieve optimality across the full range of sample sizes and accuracy levels.

## A.2. Comparing with (Jin & Sidford, 2020)

(Jin & Sidford, 2020) provides a unified framework for finding an expected $\epsilon$-optimal policy for both infinite DMDPs and AMDPs. Similar to our work, their approach is based on solving minimax problem (4) iteratively using stochastic mirror descent. Specifically, they design stochastic unbiased estimators for $v$-, $\mu$-sides, respectively, and iteratively update $v$ and $\mu$. Finally, for DMDPs they transform a $(1 - \gamma)\epsilon$-optimal solution for (4) $(v^{(\epsilon)}, \mu^{(\epsilon)})$, obtained from the minimax optimization process, into an $\epsilon$-optimal policy $\pi^{(\epsilon)}$. For DMDPs, their approach achieves a sample complexity bound of $\tilde{O}((1 - \gamma)^{-4} N \epsilon^{-2})$, matching the lower bound up to a $(1 - \gamma)^{-1}$ factor. In comparison, our work studies a different problem that we go a further step to see whether predictions of the transition matrix $\hat{P}$ can enhance the process of computing an (approximately) optimal policy and improve the sample complexity bound. We make several improvements on (Jin & Sidford, 2020)'s approach to make it adaptive into our setting.

First, we construct additional *predicted* gradients for the $\mu$ side for time $t + 1$ in (13), facilitating the update of $\mu_t$ at time $t$. This allows us to benefit from the power of optimistic mirror descent, enhancing the algorithm performance.

Second, we carefully design our learning rates to eliminate the dependence on the prediction error Dist and the desired accuracy level $\epsilon$. In (Jin & Sidford, 2020), the learning rates are defined as

$$\eta_t^v = \frac{\epsilon}{8}, \qquad \eta_t^\mu = \frac{\epsilon}{36((1 - \gamma)^{-2} + 1)N},$$

which rely on the value of $\epsilon$. If we were to construct our learning rates similarly, the learning rates should be

$$\eta_t^v = O(\epsilon), \qquad \eta_t^\mu = O\left(\frac{\epsilon}{N\gamma^2(1-\gamma)^{-2}\min\{1, \text{Dist}^2\}}\right),$$

which depend on both $\epsilon$ and prediction error Dist. To address this, we adopt a parameter-free approach using adaptive learning rates, relaxing the need to know these two values.

Third, we provide a slightly different approach to construct the stochastic estimators for the gradients on the $\boldsymbol{\mu}$-side. Our method includes a simple variance reduction technique to control the variance of the stochastic estimators, shrinking them as the sample size increases. This ensures better control in error analysis.

Altogether, while our work is inspired by the minimax approach proposed by (Jin & Sidford, 2020), we introduce essential and novel improvements over (Jin & Sidford, 2020) to incorporate the black-box predictions into the process of computing (approximately) optimal policy.

### A.3. Comparing with (Online) Algorithm with Advice (Prediction)

Regarding the prediction model, our work aligns with the research stream of (Online) Algorithm with Black-box Advice (Prediction) (Mitzenmacher & Vassilvitskii, 2022; Purohit et al., 2018). This line of research focuses on improving algorithm performance in specific problem settings by incorporating black-box advice (predictions) before the decision-making process. (Mitzenmacher & Vassilvitskii, 2022; Purohit et al., 2018) define the notions of "robustness" and "consistency", where performance metrics are based on balancing these two aspects in terms of competitive ratio. Various specific online problems have been explored, including caching (Lykouris & Vassilvitskii, 2021; Rohatgi, 2020), online resource allocation (Jiang et al., 2020; Balseiro et al., 2022; Golrezaei et al., 2023), online matching (Jin & Ma, 2022), online secretary (Antoniadis et al., 2020; Dütting et al., 2021), and convex optimization (Christianson et al., 2022). For more references, see `https://algorithms-with-predictions.github.io/`. In comparison, our work is the first to investigate leveraging predictions to enhance learning MDPs. Moreover, unlike these studies, our work focuses on sample complexity bound rather than competitive ratio-based metrics.

Another related line of research considers online optimization with predictions, aiming to decrease regret with methods like optimistic mirror descent, where the predictions are provided sequentially. Full feedback models are studied in (Rakhlin & Sridharan, 2013a;b; Jadbabaie et al., 2015), the multi-armed bandit is studied in (Wei & Luo, 2018), and the contextual bandit is studied in (Wei et al., 2020). We draw inspiration from algorithmic techniques from this line of research, such as optimistic mirror descent, to effectively incorporate black-box predictions into our minimax optimization framework. However, we emphasize that these methods cannot be directly applied to our approach. Unlike dynamic predictions, we rely on a fixed and single prediction $\hat{P}$ provided in advance of the decision-making process. Furthermore, in our algorithm, the feedback at each time step is stochastic and unbiased full feedback, differing from deterministic full feedback (Rakhlin & Sridharan, 2013a;b; Jadbabaie et al., 2015) or bandit feedback (Wei & Luo, 2018; Wei et al., 2020), necessitating a distinct analytical framework.

### A.4. Discussions on Future Directions

There are several interesting future directions.

**Extensions to other MDP or RL models** It is interesting to investigate other MDP or RL models with predictions on the transition matrix, such as average-reward MDPs (AMDPs), online tabular finite-horizon episodic MDPs and constrained MDPs. Furthermore, our algorithmic insights can inspire scalable approximations in future works, for example, the gradient nature of our approach makes it natural to combine function approximation techniques, suitable for linear MDP and other large-scale applications, even though new optimization formulations are required. Thus, it is also interesting to applying the idea in a much more complex RL environment.

**Other Forms of Predictions** Another intriguing direction is exploring whether other forms of predictions can enhance the process of solving MDPs. For example, (Golowich & Moitra, 2022) examines online tabular finite-horizon episodic MDP with prediction on Q-value, and (Li et al., 2024) considers single-trajectory time-varying MDPs with machine-learned prediction. Additionally, since such advice or predictions often originate from historical observations of other MDP models, such as robotics, it is interesting to see whether these possibly biased observations or datasets can be directly to use to enhance algorithm performance and improve sample complexity bound in solving MDPs. Moreover, it is promising to

impose additional structure on the predictions, such as the distillation assumption in (Golowich & Moitra, 2022). Such combination can improve theoretical and empirical performance.

# B. Proof for Section 3

## B.1. Notations and Auxiliary Results

**Notations** Consider a fixed algorithm ALG and an DMDP instance $\mathcal{M}$ with transition matrix P. We denote $\rho_{\mathcal{M},\text{ALG}}$ be the joint probability distribution on $\{(i_t, a_t, j_t)\}_{t=1}^T$ (where $j_t \sim p(\cdot|i_t, a_t)$), the trajectory under algorithm ALG on instance $\mathcal{M}$ with $T$ samples. For a $\sigma(\{(i_t, a_t, j_t)\}_{t=1}^T)-$measurable event $A$, we denote $\text{Pr}_{\mathcal{M},\text{ALG}}(A)$ be the probability of $A$ under $\rho_{\mathcal{M},\text{ALG}}$. For a $\sigma(\{(i_t, a_t, j_t)\}_{t=1}^T)-$measurable random variable $X$, we denote $\mathbb{E}_{\mathcal{M},\text{ALG}}[X]$ as the expectation of $X$ under $\rho_{\mathcal{M},\text{ALG}}$. We denote $N_T(i, a)$ be the number of samples on state-action pair $(i, a)$ during the whole sampling horizon, satisfying $\sum_{(i,a)\in\mathcal{N}} N_T(i, a) = T$. For a fixed policy ALG, for simplicity, we abbreviate $\text{Pr}_{\mathcal{M},\text{ALG}}(\cdot), \mathbb{E}_{\mathcal{M},\text{ALG}}[\cdot]$ as $\text{Pr}_{\mathcal{M}}(\cdot), \mathbb{E}_{\mathcal{M}}[\cdot]$, respectively.

**Auxiliary Results** Now we provide some auxiliary results from (Lattimore & Szepesvári, 2020).

**Theorem B.1.** *(Bretagnolle–Huber inequality, Theorem 14.2 in (Lattimore & Szepesvári, 2020)) Let $\mathbb{P}$, $\mathbb{Q}$ be probability measures on $(\Omega, \mathcal{F})$, and let $A \in \mathcal{F}$ be an arbitrary event. Then*

$$\Pr_{\mathbb{P}}(A) + \Pr_{\mathbb{Q}}(A^c) \geq \frac{1}{2} \exp\left(-KL(\mathbb{P}, \mathbb{Q})\right).$$

**Theorem B.2.** *(Divergence decomposition, Lemma 15.1 in (Lattimore & Szepesvári, 2020)) Consider two instances $\mathcal{M}_1$, $\mathcal{M}_2$ that share the same state space, same action space for each state, same number of samples $T$. For any sampling algorithm ALG,*

$$KL(\Pr_{\mathcal{M}_1}, \Pr_{\mathcal{M}_2}) = \sum_{(i,a)\in\mathcal{N}} \mathbb{E}_{\mathcal{M}_1}[N_T(i, a)] \cdot KL(\Pr_{\mathcal{M}_1,(i,a)}, \Pr_{\mathcal{M}_2,(i,a)}).$$

## B.2. Proof for Theorem 3.2

This proof is inspried by the lower bound example provided by (Gheshlaghi Azar et al., 2013). Throughout the proof we assume $\gamma \geq \frac{1}{3}$.

We define model class $\mathcal{I}_{m,n}$, with $m, n \geq 1$ and $mn > 1$: For any DMDP model $\mathcal{M} \in \mathcal{I}_{m,n}$,

- State $\mathcal{S}$: $\mathcal{S}$ includes $m$ "start" states $\{i_k^{(\text{st})}\}_{k\in[m]}$, $mn$ "middle" states $\{i_{k,\ell}^{(\text{mi})}\}_{k\in[m],\ell\in[n]}$, and $mn$ "end" states $\{i_{k,\ell}^{(\text{en})}\}_{k\in[m],\ell\in[n]}$.

- Action $\mathcal{A}$:
    - "start" states $\{i_k^{(\text{st})}\}_{k\in[m]}$: For each $k \in [m]$, the state $i_k^{(\text{st})}$ can take $n$ actions $\{a_\ell\}_{\ell\in[n]}$. For each state action pair $(i_k^{(\text{st})}, a_\ell)$, it receives reward 0 and transits to state $i_{k,\ell}^{(\text{mi})}$ with probability 1.
    - "middle" states $\{i_{k,\ell}^{(\text{mi})}\}_{k\in[m],\ell\in[n]}$: For each $(k, \ell)$, the state $i_{k,\ell}^{(\text{mi})}$ can only take 1 action $a^{(\text{mi})}$. After taking this action, it receives reward 1 with probability 1. It transits to itself with probability $p_{\mathcal{M}}(k, \ell) \in [0, 1]$, and with probability $1 - p_{\mathcal{M}}(k, \ell)$ transits to $i_{k,\ell}^{(\text{en})}$.
    - "end" states $\{i_{k,\ell}^{(\text{en})}\}_{k\in[m],\ell\in[n]}$: For each $(k, \ell)$, the state $i_{k,\ell}^{(\text{en})}$ can only take 1 action $a^{(\text{en})}$, receiving reward 0 and transiting to itself with probablity 1.

Note that the total number of state-action pairs $N = 3mn$. We define instance $\mathcal{M}_0 \in \mathcal{I}_{m,n}$ with transition matrix $P_0$ and prediction matrix $\hat{P} = P_0$, which is defined as follows:

$$p_{\mathcal{M}_0}(k, \ell) = \begin{cases} p_0 + \Delta & (k, \ell) = (1, 1), \\ p_0 & \text{otherwise.} \end{cases}$$

where $p_0 = \frac{4\gamma - 1}{3\gamma}$, and $\Delta$ is to be determined later. For this model, the optimal value vector is

$$v^{*,(\text{en})}_{\mathcal{M}_0,k,\ell} = 0, \qquad v^{*,(\text{mi})}_{\mathcal{M}_0,k,\ell} = \begin{cases} \frac{1}{1-\gamma(p_0+\Delta)} & (k,\ell) = (1,1), \\ \frac{1}{1-\gamma p_0} & \text{otherwise,} \end{cases} \qquad v^{*,(\text{st})}_{\mathcal{M}_0,k} = \gamma \max_{\ell \in [n]} v^{*,(\text{mi})}_{\mathcal{M}_0,k,\ell}.$$

For simplicity, we further assume that for any algorithm ALG that access to the knowledge of transition for "start" and "end" states, and only does not know about the transition probability for "middle" states $\{p_{\mathcal{M}_0}(k,\ell)\}_{k\in[m],\ell\in[n]}$. Now, we show that for any ALG, if it can be $(\epsilon, \delta)$-smart ($\epsilon \in (0, \frac{1}{40}(1-\gamma)^{-1})$, $\delta \in (0, 0.24)$ on $\mathcal{M}_0$ with the help of prediction $\hat{\mathsf{P}}$ with $T$ satisfying

$$T \leq \frac{1}{300}(1-\gamma)^{-3}(mn-1)\epsilon^{-2}\ln\left(\frac{1}{4.1\delta}\right),$$

then there exists $\mathcal{M}' \in \mathcal{I}_{m,n}$ such that ALG cannot be $(\epsilon, \delta)$-smart on $\mathcal{M}'$ with a same $T$ and same prediction $\hat{\mathsf{P}}$.

We start from $N_T(\cdot, \cdot)$:

$$\sum_{k\in[m],\ell\in[n]} \mathbb{E}_{\mathcal{M}_0}[N_T(i^{(\text{mi})}_{k,\ell}, a^{(\text{mi})})] = T.$$

Then there exist $(k', \ell') \neq (1, 1)$ such that

$$\mathbb{E}_{\mathcal{M}_0}[N_T(i^{(\text{mi})}_{k',\ell'}, a^{(\text{mi})})] \leq \frac{T}{mn-1}.$$

Now, we construct $\mathcal{M}' \in \mathcal{I}_{m,n}$ as follows:

$$p_{\mathcal{M}'}(k,\ell) = \begin{cases} p_0 + \Delta & (k,\ell) = (1,1), \\ p_0 + 2\Delta & (k,\ell) = (k',\ell'), \\ p_0 & \text{otherwise.} \end{cases}$$

Let $\Delta = \frac{5}{3} \cdot \frac{(1-\gamma)^2}{\gamma}\epsilon$, then we have

$$\Pr_{\mathcal{M}_0,T}\left(\left\|v^{\hat{\pi}_T} - v^*_{\mathcal{M}_0}\right\|_\infty > \epsilon\right) + \Pr_{\mathcal{M}',T}\left(\left\|v^{\hat{\pi}_T} - v^*_{\mathcal{M}'}\right\|_\infty > \epsilon\right)$$

$$\geq \Pr_{\mathcal{M}_0,T}\left(\left\|v^{\hat{\pi}_T} - v^*_{\mathcal{M}_0}\right\|_\infty > \epsilon\right) + \Pr_{\mathcal{M}',T}\left(\left\|v^{\hat{\pi}_T} - v^*_{\mathcal{M}_0}\right\|_\infty \leq \epsilon\right) \tag{17a}$$

$$\geq \frac{1}{2}\exp\left[-\text{KL}\left(\Pr_{\mathcal{M}_0,T}, \Pr_{\mathcal{M}',T}\right)\right] \tag{17b}$$

$$= \frac{1}{2}\exp\left[-\mathbb{E}_{\mathcal{M}_0}[N_T(i^{(\text{mi})}_{k',\ell'}, a^{(\text{mi})})] \cdot \text{KL}\left(\text{Bern}\left(p_0\right), \text{Bern}\left(p_0 + 2\Delta\right)\right)\right] \tag{17c}$$

$$\geq \frac{1}{2}\exp\left[-\frac{1}{300} \cdot \frac{(1-\gamma)^{-3}(mn-1)\epsilon^{-2}}{mn-1} \cdot \frac{4\Delta^2}{p_0(1-p_0)} \cdot \ln\left(\frac{1}{4.1\delta}\right)\right] \tag{17d}$$

$$\geq \frac{1}{2}\exp\left[-\frac{1}{300} \cdot \frac{(1-\gamma)^{-3}(mn-1)\epsilon^{-2}}{mn-1} \cdot \frac{12\Delta^2}{1-p_0} \cdot \ln\left(\frac{1}{4.1\delta}\right)\right] \tag{17e}$$

$$\geq \frac{1}{2}\exp\left(-\ln\left(\frac{1}{4.1\delta}\right)\right) > 2\delta. \tag{17f}$$

where (17a) comes from the following: by $\epsilon < \frac{1}{40}(1-\gamma)^{-1}$, we have $2\Delta \leq \frac{1-\gamma}{12\gamma}$, then

$$p_0 + 2\Delta \leq \frac{4\gamma - 1}{3\gamma} + \frac{1-\gamma}{12\gamma} = \frac{5\gamma - 1}{4\gamma}, \qquad \Rightarrow \qquad \frac{1}{1-\gamma(p_0 + 2\Delta)} \geq \frac{4}{5}\frac{1}{1-\gamma}.$$

We can also derive $\frac{1}{1-\gamma p_0} = \frac{3}{4}\frac{1}{1-\gamma}$. Therefore,

$$\frac{1}{1-\gamma(p_0+2\Delta)} - \frac{1}{1-\gamma p_0} = \frac{2\gamma\Delta}{(1-\gamma(p_0+2\Delta))(1-\gamma p_0)} \geq 2\gamma \cdot \frac{5}{3}\frac{(1-\gamma)^2\epsilon}{\gamma} \cdot \frac{4}{5}\frac{1}{1-\gamma} \cdot \frac{3}{4}\frac{1}{1-\gamma} = 2\epsilon.$$

This implies that the subset of events that perform "bad" at $\mathcal{M}'$ ($\|v^{\hat{\pi}_T} - v^*_{\mathcal{M}_0}\|_\infty \le \epsilon$) is a subset of events that perform "well" at $\mathcal{M}_0$ ($\|v^{\hat{\pi}_T} - v^*_{\mathcal{M}'}\|_\infty > \epsilon$). Or in other words, here the difference between $\mathcal{M}_0$ and $\mathcal{M}'$ is large enough that when it can perform well or either of the instances and can not on both. (17b) comes from Bretagnolle–Huber inequality (Theorem B.1). (17c) comes from Divergence Decomposition (Theorem B.2). (17d) comes from following:

$$
\begin{aligned}
\text{KL}\left(\text{Bern}\left(p_0\right), \text{Bern}\left(p_0 + 2\Delta\right)\right) &= p_0 \ln\left(\frac{p_0}{p_0 + 2\Delta}\right) + (1 - p_0)\ln\left(\frac{1 - p_0}{1 - p_0 - 2\Delta}\right) \\
&= -p_0 \ln\left(\frac{p_0 + 2\Delta}{p_0}\right) - (1 - p_0)\ln\left(\frac{1 - p_0 - 2\Delta}{1 - p_0}\right) \\
&\le p_0\left(\frac{4\Delta^2}{p_0} - \frac{4\Delta}{p_0}\right) + (1 - p_0)\left(\frac{4\Delta^2}{(1 - p_0)^2} + \frac{2\Delta}{1 - p_0}\right) \\
&\le \frac{4\Delta^2}{p_0} + \frac{4\Delta^2}{1 - p_0} = \frac{4\Delta^2}{p_0(1 - p_0)},
\end{aligned}
$$

where the first inequality comes from the following two facts:

$$
\ln(1 + x) \ge x - \frac{x^2}{2}, \qquad \forall x \in [0, 1], \qquad \ln(1 - x) \ge -x - x^2, \qquad \forall x \in \left[0, \frac{1}{2}\right].
$$

It is also straightforward to verify that

$$
0 < \frac{2\Delta}{p_0} \le \frac{\frac{1-\gamma}{12\gamma}}{\frac{4\gamma-1}{3\gamma}} \le \frac{1}{4} < 1, \qquad 0 < \frac{2\Delta}{1 - p_0} \le \frac{\frac{1-\gamma}{12\gamma}}{1 - \frac{4\gamma-1}{3\gamma}} \le \frac{1}{8} < \frac{1}{2}.
$$

(17e) comes from the fact that $\gamma \ge \frac{1}{3}$ so $p_0 = \frac{4\gamma-1}{3\gamma} \ge \frac{1}{3}$. (17f) comes from following:

$$
\frac{(1-\gamma)^{-3}(mn-1)\epsilon^{-2}}{mn-1} \cdot \frac{12\Delta^2}{1 - p_0} = \frac{(1-\gamma)^{-3}(mn-1)\epsilon^{-2}}{mn-1} \cdot \frac{12}{1 - \frac{4\gamma-1}{3\gamma}} \cdot \left(\frac{5}{3}\frac{(1-\gamma)^2}{\gamma}\epsilon\right)^2 = 100 \cdot \frac{1}{\gamma} \le 300.
$$

According to the assumption that ALG is $(\epsilon, \delta)$-*smart* on $\mathcal{M}_0$ with the help of prediction $\hat{P}$, then we have

$$
\Pr_{\mathcal{M}_0, T}\left(\|v^{\hat{\pi}_T} - v^*_{\mathcal{M}_0}\|_\infty > \epsilon\right) < \delta.
$$

Therefore by (17),

$$
\Pr_{\mathcal{M}', T}\left(\|v^{\hat{\pi}_T} - v^*_{\mathcal{M}'}\|_\infty > \epsilon\right) > 2\delta - \Pr_{\mathcal{M}_0, T}\left(\|v^{\hat{\pi}_T} - v^*_{\mathcal{M}_0}\|_\infty > \epsilon\right) > \delta.
$$

This implies that ALG is not $(\epsilon, \delta)$-*smart* on $\mathcal{M}'$. Altogether, the Theorem is proved.

### B.3. More Discussions on the Proof for Theorem 3.2

We provide further discussions on the instances $\mathcal{M}_0$, $\mathcal{M}'$. For instance $\mathcal{M}_0$, the state-action pair $(i^{(\text{mi})}_{1,1}, a^{(\text{mi})})$ is the most "valueable" pair, meaning that focusing on this pair as frequently as possible leads to the optimal value function. In contrast, for instance $\mathcal{M}'$, the most "valueable" pair is $i^{(\text{mi})}_{k',\ell'}, a^{(\text{mi})}$, where $(k', \ell') \ne (1, 1)$. Note that the ALG is provided with the same prediction $\hat{P}$ $(= P_0)$. Thus, while $\hat{P}$ provides useful information for ALG for learning $\mathcal{M}_0$, it is misleading for learning $\mathcal{M}'$. As a result, no policy can simultaneously confirm that (1) $\hat{P}$ is useful for learning $\mathcal{M}_0$, and (2) $\hat{P}$ is misleading for learning $\mathcal{M}'$ and should be ignored entirely. This reasoning ultimately leads to Theorem 3.2.

## C. Proof in Section 4

### C.1. Proof for Lemma 4.1, 4.2

We rephrase Lemma 4.1, 4.2 as Lemma C.1, Lemma C.2, respectively, and provide their proofs.

**Lemma C.1.** *(Lemma 3 in (Jin & Sidford, 2020)) The stochastic gradient $\tilde{g}^v_t$ for $v$ satisfies $\mathbb{E}[\tilde{g}^v_t] = (1-\gamma)q + \mu^\top_t(\gamma P - \hat{I}) = g^v(v_t, \mu_t)$ and $\mathbb{E}[\|\tilde{g}^v_t\|^2_2] \le 2.$*

*Proof.* We can directly compute

$$\mathbb{E}[\tilde{g}_t^v] = (1-\gamma)q + \sum_{i,a,j} \mu_{t,(i,a)} p(j|i,a)(\gamma e_j - e_i) = (1-\gamma)q + \mu^\top(\gamma P - \hat{I}).$$

By definition, it is trivial to check $\|\tilde{g}_t^v\|_2^2 \le (1-\gamma)^2 + \gamma^2 + 1 \le 2$ with probability 1. $\qquad \square$

**Lemma C.2.** *The stochastic gradient $\tilde{g}_t^\mu$ for $v$ satisfies $\mathbb{E}[\tilde{g}_t^\mu] = (\hat{I} - \gamma P)v_t - r = g^\mu(v_t, \mu_t)$ and $\mathbb{E}[\|\tilde{g}_t^\mu - g^\mu(v, \mu)\|_\infty^2] \le 9N^2(1-\gamma)^{-2}/t$.*

*Proof.* The proof largely follows the proof for Lemma 4 in (Jin & Sidford, 2020). Denote $\tilde{g}_t^{\mu,\ell} = N(v_{t,i_\ell} - \gamma v_{t,j_\ell} - r_{i_\ell,a_\ell})e_{i_\ell,a_\ell}$, where $(i_\ell, a_\ell, j_\ell)$ be the $\ell-$th pair sampled. Then $\tilde{g}_t^\mu = \frac{1}{t}\sum_{\ell=1}^t \tilde{g}_t^{\mu,\ell}$. For each $\tilde{g}_t^{\mu,\ell}$,

$$\mathbb{E}[\tilde{g}_t^{\mu,\ell}] = \sum_{i,a}\sum_j p(j|i,a)(v_i - \gamma v_j - r_{i,a})e_{i,a} = (\hat{I} - \gamma \hat{P})v - r = g^\mu(v, \mu),$$

and $\|\tilde{g}_t^{\mu,\ell}\|_\infty \le 3N(1-\gamma)^{-1}$ with probability 1. Hence, $\mathbb{E}[\tilde{g}_t^\mu] = \frac{1}{t}\sum_{\ell=1}^t \mathbb{E}[\tilde{g}_t^{\mu,\ell}] = g^\mu(v, \mu)$, and

$$\mathbb{E}[\|\tilde{g}_t^v - g^\mu(v, \mu)\|_\infty^2] \le \frac{1}{t}\max_{\ell \in [t]} \mathbb{E}[\|\tilde{g}_t^\mu - g^\mu(v, \mu)\|_\infty^2] \le \frac{9N^2(1-\gamma)^{-2}}{t}.$$

$\qquad \square$

## C.2. Proof for Lemma 4.4

We derive a stronger version for Lemma 4.4. We will show that for any $v \in \mathcal{V}$, $\mu \in \mathcal{U}$,

$$\mathbb{E}[f(\bar{v}, \mu) - f(v, \bar{\mu})]$$
$$\le 3\left(\sqrt{|\mathcal{S}|}(1-\gamma)^{-1} + \sqrt{N} \cdot \gamma(1-\gamma)^{-1} \cdot \min\{1, \text{Dist}\}\right) \cdot \sqrt{\frac{1}{T}} + 9\sqrt{2}N(1-\gamma)^{-1}\frac{\ln(T)}{T}.$$

This is because
$$\mathbb{E}[\text{GAP}(\bar{v}, \bar{\mu})] = \mathbb{E}[\max_{\mu'} f(\bar{v}, \mu') - \min_{v'} f(v', \bar{\mu})] = \mathbb{E}[f(\bar{v}, \bar{\mu}') - f(\bar{v}', \bar{\mu})],$$

where $\bar{\mu}' \in \arg\max_{\mu \in \mathcal{U}} f(\bar{v}, \mu)$ and $\bar{v}' \in \arg\max_{v \in \mathcal{V}} f(v, \bar{\mu})$. Denote $g^v(v, \mu) = \nabla_v f(v, \mu) = (1-\gamma)q + \mu^\top(\gamma P - \hat{I})$, $g^\mu(v, \mu) = -\nabla_\mu f(v, \mu) = (\hat{I} - \gamma P)v - r$. Denote

$$\hat{f}_t^{(v)}(v, \mu) = (\tilde{g}_t^v)^\top v + \mu^\top r.$$

We remark that by the construction of $\tilde{g}^v$, $\tilde{g}^v$ is a function of $\mu$. By Lemma C.1, we have $\mathbb{E}[\tilde{g}^v(v, \mu)] = (1-\gamma)q + \mu^\top(\gamma P - \hat{I}) = g^v(v, \mu)$. Similarly, denote

$$\hat{f}_t^{(\mu)}(v, \mu) = (1-\gamma)q^\top v - \mu^\top \tilde{g}_t^\mu.$$

$\tilde{g}^\mu$ is a function of $v$. By Lemma C.2, we have $\mathbb{E}[\tilde{g}^\mu(v, \mu)] = (\hat{I} - \gamma P)v - r = g^\mu(v, \mu)$.

Now, we decompose $\mathbb{E}[f(\bar{v}, \mu) - f(v, \bar{\mu})]$ as follows: For any $v \in \mathcal{V}$, $\mu \in \mathcal{U}$,

$$\mathbb{E}\left[f(\bar{v}, \mu) - f(v, \bar{\mu})\right]$$

$$= \mathbb{E}\left[\frac{1}{T}\sum_{t=1}^T f(v_t, \mu) - \frac{1}{T}\sum_{t=1}^T f(v, \mu_t)\right] \tag{18a}$$

$$= \frac{1}{T}\mathbb{E}\left[\sum_{t=1}^T f(v_t, \mu) - \sum_{t=1}^T f(v_t, \mu_t)\right] + \frac{1}{T}\mathbb{E}\left[\sum_{t=1}^T f(v_t, \mu_t) - \sum_{t=1}^T f(v, \mu_t)\right]$$

$$\le \frac{1}{T}\mathbb{E}\left[\sum_{t=1}^T \hat{f}_t^{(\mu)}(v_t, \mu) - \sum_{t=1}^T \hat{f}_t^{(\mu)}(v_t, \mu_t)\right] + \frac{1}{T}\mathbb{E}\left[\sum_{t=1}^T \hat{f}_t^{(v)}(v_t, \mu_t) - \sum_{t=1}^T \hat{f}_t^{(v)}(v, \mu_t)\right]. \tag{18b}$$

(18a) comes from the bilinear structure of $f(\cdot, \cdot)$. The first term in (18b) comes from the following:

$$
\begin{aligned}
&\mathbb{E}\left[\sum_{t=1}^{T} f(\boldsymbol{v}_t, \boldsymbol{\mu}) - \sum_{t=1}^{T} f(\boldsymbol{v}_t, \boldsymbol{\mu}_t)\right] \\
=&\mathbb{E}\left[\sum_{t=1}^{T} f(\boldsymbol{v}_t, \boldsymbol{\mu}) - \sum_{t=1}^{T} \hat{f}_t^{(\boldsymbol{\mu})}(\boldsymbol{v}_t, \boldsymbol{\mu}) + \sum_{t=1}^{T} \hat{f}_t^{(\boldsymbol{\mu})}(\boldsymbol{v}_t, \boldsymbol{\mu}) - \sum_{t=1}^{T} \hat{f}_t^{(\boldsymbol{\mu})}(\boldsymbol{v}_t, \boldsymbol{\mu}_t) + \sum_{t=1}^{T} \hat{f}_t^{(\boldsymbol{\mu})}(\boldsymbol{v}_t, \boldsymbol{\mu}_t) - \sum_{t=1}^{T} f(\boldsymbol{v}_t, \boldsymbol{\mu}_t)\right] \\
=&\mathbb{E}\left[\sum_{t=1}^{T} \hat{f}_t^{(\boldsymbol{\mu})}(\boldsymbol{v}_t, \boldsymbol{\mu}) - \sum_{t=1}^{T} \hat{f}_t^{(\boldsymbol{\mu})}(\boldsymbol{v}_t, \boldsymbol{\mu}_t)\right] + \mathbb{E}\left[\sum_{t=1}^{T} (\tilde{\boldsymbol{g}}_t^{\boldsymbol{\mu}} - \boldsymbol{g}^{\boldsymbol{\mu}}(\boldsymbol{v}_t, \boldsymbol{\mu}))^{\top} \boldsymbol{\mu}\right] + \mathbb{E}\left[\sum_{t=1}^{T} (\boldsymbol{g}^{\boldsymbol{\mu}}(\boldsymbol{v}_t, \boldsymbol{\mu}_t) - \tilde{\boldsymbol{g}}_t^{\boldsymbol{\mu}})^{\top} \boldsymbol{\mu}_t\right] \\
=&\mathbb{E}\left[\sum_{t=1}^{T} \hat{f}_t^{(\boldsymbol{\mu})}(\boldsymbol{v}_t, \boldsymbol{\mu}) - \sum_{t=1}^{T} \hat{f}_t^{(\boldsymbol{\mu})}(\boldsymbol{v}_t, \boldsymbol{\mu}_t)\right].
\end{aligned}
$$

The last inequality comes from fact that $\boldsymbol{g}^{\boldsymbol{\mu}}(\boldsymbol{v}, \boldsymbol{\mu}) = (\hat{\mathrm{I}} - \gamma\hat{\mathrm{P}})\boldsymbol{v} - \mathrm{r}$ does not depend on $\boldsymbol{\mu}$, and the conditional expectation that $\forall t, \boldsymbol{\mu}' \in \mathcal{U}$,

$$
\mathbb{E}\left[(\tilde{\boldsymbol{g}}_t^{\boldsymbol{\mu}} - \boldsymbol{g}^{\boldsymbol{\mu}}(\boldsymbol{v}_t, \boldsymbol{\mu}))^{\top} \boldsymbol{\mu}'\right] = \mathbb{E}\left[\mathbb{E}\left[(\tilde{\boldsymbol{g}}_t^{\boldsymbol{\mu}} - \boldsymbol{g}^{\boldsymbol{\mu}}(\boldsymbol{v}_t, \boldsymbol{\mu}))^{\top} \boldsymbol{\mu}' | \sigma(\{(i_k, a_k, j_k)\}_{k=1}^{t-1})\right]\right] = 0.
$$

Similarly, the second term in (18b) comes from the following:

$$
\begin{aligned}
&\mathbb{E}\left[\sum_{t=1}^{T} f(\boldsymbol{v}_t, \boldsymbol{\mu}_t) - \sum_{t=1}^{T} f(\boldsymbol{v}, \boldsymbol{\mu}_t)\right] \\
=&\mathbb{E}\left[\sum_{t=1}^{T} f(\boldsymbol{v}_t, \boldsymbol{\mu}_t) - \sum_{t=1}^{T} \hat{f}_t^{(\boldsymbol{v})}(\boldsymbol{v}_t, \boldsymbol{\mu}_t) + \sum_{t=1}^{T} \hat{f}_t^{(\boldsymbol{v})}(\boldsymbol{v}_t, \boldsymbol{\mu}_t) - \sum_{t=1}^{T} \hat{f}_t^{(\boldsymbol{v})}(\boldsymbol{v}, \boldsymbol{\mu}_t) + \sum_{t=1}^{T} \hat{f}_t^{(\boldsymbol{v})}(\boldsymbol{v}, \boldsymbol{\mu}_t) - \sum_{t=1}^{T} f(\boldsymbol{v}, \boldsymbol{\mu}_t)\right] \\
=&\mathbb{E}\left[\sum_{t=1}^{T} \hat{f}_t^{(\boldsymbol{v})}(\boldsymbol{v}_t, \boldsymbol{\mu}_t) - \sum_{t=1}^{T} \hat{f}_t^{(\boldsymbol{v})}(\boldsymbol{v}, \boldsymbol{\mu}_t)\right] + \mathbb{E}\left[\sum_{t=1}^{T} (\boldsymbol{g}^{\boldsymbol{v}}(\boldsymbol{v}_t, \boldsymbol{\mu}_t) - \tilde{\boldsymbol{g}}_t^{\boldsymbol{v}})^{\top} \boldsymbol{v}_t\right] + \mathbb{E}\left[\sum_{t=1}^{T} (\tilde{\boldsymbol{g}}_t^{\boldsymbol{v}} - \boldsymbol{g}^{\boldsymbol{v}}(\boldsymbol{v}, \boldsymbol{\mu}_t))^{\top} \boldsymbol{v}\right] \\
=&\mathbb{E}\left[\sum_{t=1}^{T} \hat{f}_t^{(\boldsymbol{v})}(\boldsymbol{v}_t, \boldsymbol{\mu}_t) - \sum_{t=1}^{T} \hat{f}_t^{(\boldsymbol{v})}(\boldsymbol{v}, \boldsymbol{\mu}_t)\right].
\end{aligned}
$$

Now we analyze the two terms of (18b), respectively.

**Analysis for $\mathbb{E}\left[\sum_{t=1}^{T} f(\boldsymbol{v}_t, \boldsymbol{\mu}) - \sum_{t=1}^{T} f(\boldsymbol{v}_t, \boldsymbol{\mu}_t)\right]$:** Notice that the update fule for $\boldsymbol{\mu}_t$ in (10) is equivalent to the following optimistic online mirror descent update rule (Algorithm 6.4 in (Orabona, 2019)):

$$
\boldsymbol{\mu}_{t+1} = \underset{\boldsymbol{\mu} \in \mathcal{U}}{\arg\min}\left\{(\tilde{\boldsymbol{g}}_t^{\boldsymbol{\mu}} - \bar{\boldsymbol{g}}_t^{\boldsymbol{\mu}} + \bar{\boldsymbol{g}}_{t+1}^{\boldsymbol{\mu}})^{\top} \boldsymbol{\mu} + \frac{1}{\eta_t^{\boldsymbol{\mu}}} B_{\psi}(\boldsymbol{\mu}; \boldsymbol{\mu}_t)\right\}.
$$

where $B_\psi(\boldsymbol{z}; \boldsymbol{y}) = \sum_{i \in [N]} z_i \ln \frac{z_i}{y_i}$. Then we have

$$\sum_{t=1}^{T} \hat{f}_t^{(\boldsymbol{\mu})}(\boldsymbol{v}_t, \boldsymbol{\mu}) - \sum_{t=1}^{T} \hat{f}_t^{(\boldsymbol{\mu})}(\boldsymbol{v}_t, \boldsymbol{\mu}_t) = \sum_{t=1}^{T} (\tilde{\boldsymbol{g}}^{\boldsymbol{\mu}}(\boldsymbol{v}_t, \boldsymbol{\mu}_t))^\top \boldsymbol{\mu}_t - \sum_{t=1}^{T} (\tilde{\boldsymbol{g}}^{\boldsymbol{\mu}}(\boldsymbol{v}, \boldsymbol{\mu}_t))^\top \boldsymbol{\mu}$$

$$= \sum_{t=1}^{T} (\tilde{\boldsymbol{g}}^{\boldsymbol{\mu}}(\boldsymbol{v}_t, \boldsymbol{\mu}_t))^\top \boldsymbol{\mu}_t - \sum_{t=1}^{T} (\tilde{\boldsymbol{g}}^{\boldsymbol{\mu}}(\boldsymbol{v}_t, \boldsymbol{\mu}_t))^\top \boldsymbol{\mu} \tag{19a}$$

$$\leq \frac{\ln(N)}{\eta_T^\mu} + \frac{1}{2} \sum_{t=1}^{T} \eta_t^\mu \|\tilde{\boldsymbol{g}}_t^{\boldsymbol{\mu}} - \bar{\boldsymbol{g}}_t^{\boldsymbol{\mu}}\|_\infty^2 \tag{19b}$$

$$= \sqrt{2} \cdot \sqrt{\ln(N)} \cdot \sqrt{\sum_{t=1}^{T} \|\tilde{\boldsymbol{g}}_t^{\boldsymbol{\mu}} - \bar{\boldsymbol{g}}_t^{\boldsymbol{\mu}}\|_\infty^2}$$

$$+ \frac{\sqrt{2}}{4} \cdot \sqrt{\ln(N)} \cdot \sum_{t=1}^{T} \frac{\|\tilde{\boldsymbol{g}}_t^{\boldsymbol{\mu}} - \bar{\boldsymbol{g}}_t^{\boldsymbol{\mu}}\|_\infty^2}{\sqrt{\sum_{i=1}^{t} \|\tilde{\boldsymbol{g}}_i^{\boldsymbol{\mu}} - \bar{\boldsymbol{g}}_i^{\boldsymbol{\mu}}\|_\infty^2}} \tag{19c}$$

$$\leq \frac{3}{2} \sqrt{2} \cdot \sqrt{\ln(N)} \cdot \sqrt{\sum_{t=1}^{T} \|\tilde{\boldsymbol{g}}_t^{\boldsymbol{\mu}} - \bar{\boldsymbol{g}}_t^{\boldsymbol{\mu}}\|_\infty^2}. \tag{19d}$$

(19a) comes from the fact that $\tilde{\boldsymbol{g}}^{\boldsymbol{\mu}}$ only depends on $\boldsymbol{v}$ based on its construction rule. (19b) comes from Theorem 6.20 in (Orabona, 2019) (we take $\ell_t(\boldsymbol{x}) = \tilde{\boldsymbol{g}}_t^{\boldsymbol{\mu}, \top} \boldsymbol{x}$, and $\bar{\boldsymbol{g}}_t$ as the prediction of the subgradient). (19c) comes from the definition of $\eta_t^\mu$. (19d) comes from Lemma 4.13 in (Orabona, 2019). Therefore,

$$\mathbb{E}\left[\sum_{t=1}^{T} \hat{f}_t^{(\boldsymbol{\mu})}(\boldsymbol{v}_t, \boldsymbol{\mu}) - \sum_{t=1}^{T} \hat{f}_t^{(\boldsymbol{\mu})}(\boldsymbol{v}_t, \boldsymbol{\mu}_t)\right]$$

$$\leq \frac{3}{2} \sqrt{2} \cdot \sqrt{\ln(N)} \cdot \sqrt{\mathbb{E}\left[\sum_{t=1}^{T} \|\tilde{\boldsymbol{g}}_t^{\boldsymbol{\mu}} - \bar{\boldsymbol{g}}_t^{\boldsymbol{\mu}}\|_\infty^2\right]} \tag{20a}$$

$$\leq 3\sqrt{\ln(N)} \sqrt{\mathbb{E}\left[\sum_{t=1}^{T} \|\tilde{\boldsymbol{g}}_t^{\boldsymbol{\mu}} - \boldsymbol{g}^{\boldsymbol{\mu}}(\boldsymbol{v}_t, \boldsymbol{\mu}_t)\|_\infty^2\right] + \mathbb{E}\left[\sum_{t=1}^{T} \|\boldsymbol{g}^{\boldsymbol{\mu}}(\boldsymbol{v}_t, \boldsymbol{\mu}_t) - \bar{\boldsymbol{g}}_t^{\boldsymbol{\mu}}\|_\infty^2\right]} \tag{20b}$$

$$= 3\sqrt{\ln(N)} \sqrt{\sum_{t=1}^{T} \mathbb{E}\left[\|\tilde{\boldsymbol{g}}_t^{\boldsymbol{\mu}} - \boldsymbol{g}^{\boldsymbol{\mu}}(\boldsymbol{v}_t, \boldsymbol{\mu}_t)\|_\infty^2\right] + \sum_{t=1}^{T} \mathbb{E}\left[\|\boldsymbol{g}^{\boldsymbol{\mu}}(\boldsymbol{v}_t, \boldsymbol{\mu}_t) - \bar{\boldsymbol{g}}_t^{\boldsymbol{\mu}}\|_\infty^2\right]}$$

$$\leq 3\sqrt{\ln(N)} \sqrt{\sum_{t=1}^{T} \frac{9N^2(1-\gamma)^{-2}}{t} + T \cdot N \cdot \gamma^2(1-\gamma)^{-2} \cdot \min\left\{1, \text{Dist}^2\right\}} \tag{20c}$$

$$\leq 3\sqrt{\ln(N)} \sqrt{18N^2(1-\gamma)^{-2} \cdot \ln(T) + T \cdot N \cdot \gamma^2(1-\gamma)^{-2} \cdot \min\left\{1, \text{Dist}^2\right\}}$$

(20a) comes from Jensen Inequality. (20b) comes from the fact that for any two vectors $\boldsymbol{a}, \boldsymbol{b}$, $\|\boldsymbol{a} + \boldsymbol{b}\|_\infty^2 \leq 2\|\boldsymbol{a}\|_\infty^2 + 2\|\boldsymbol{b}\|_\infty^2$. (20c) comes from Lemma C.2 and the following calculation:

$$\boldsymbol{g}^{\boldsymbol{\mu}}(\boldsymbol{v}_t, \boldsymbol{\mu}_t) - \bar{\boldsymbol{g}}_t^{\boldsymbol{\mu}} = (\hat{\mathrm{I}} - \gamma\mathrm{P})\boldsymbol{v}_t - \mathrm{r} - \left((\hat{\mathrm{I}} - \gamma\hat{\mathrm{P}})\boldsymbol{v}_t - \mathrm{r}\right) = \gamma(\mathrm{P} - \hat{\mathrm{P}})\boldsymbol{v}_t,$$

so

$$\mathbb{E}\left[\|\boldsymbol{g}^{\boldsymbol{\mu}}(\boldsymbol{v}_t, \boldsymbol{\mu}_t) - \bar{\boldsymbol{g}}_t^{\boldsymbol{\mu}}\|_\infty^2\right] \leq N \cdot \gamma^2(1-\gamma)^{-2} \cdot \min\left\{1, \text{Dist}^2\right\}.$$

**Analysis for** $\mathbb{E}\left[\sum_{t=1}^{T} f(\boldsymbol{v}_t, \boldsymbol{\mu}_t) - \sum_{t=1}^{T} f(\boldsymbol{v}, \boldsymbol{\mu}_t)\right]$**:** This analysis largely follows the line as the previous part. Notice that

the update rule for $\boldsymbol{v}_t$ in (8) is equivalent to the following online mirror descent (OMD) rule:

$$\boldsymbol{v}_{t+1} = \arg\min_{\boldsymbol{v}\in\mathcal{V}} \left\{ (\tilde{\boldsymbol{g}}_t^{\boldsymbol{v}})^\top \boldsymbol{v} + \frac{1}{2\eta_t}\|\boldsymbol{v}_t - \boldsymbol{v}\|_2^2 \right\}.$$

Then we have

$$\sum_{t=1}^T \hat{f}^{(\boldsymbol{v})}(\boldsymbol{v}_t, \boldsymbol{\mu}_t) - \sum_{t=1}^T \hat{f}^{(\boldsymbol{v})}(\boldsymbol{v}, \boldsymbol{\mu}_t) = \sum_{t=1}^T (\tilde{\boldsymbol{g}}^{\boldsymbol{v}}(\boldsymbol{v}_t, \boldsymbol{\mu}_t))^\top \boldsymbol{v}_t - \sum_{t=1}^T (\tilde{\boldsymbol{g}}^{\boldsymbol{v}}(\boldsymbol{v}, \boldsymbol{\mu}_t))^\top \boldsymbol{v}$$

$$= \sum_{t=1}^T (\tilde{\boldsymbol{g}}^{\boldsymbol{v}}(\boldsymbol{v}_t, \boldsymbol{\mu}_t))^\top \boldsymbol{v}_t - \sum_{t=1}^T (\tilde{\boldsymbol{g}}^{\boldsymbol{v}}(\boldsymbol{v}_t, \boldsymbol{\mu}_t))^\top \boldsymbol{v}$$

$$\leq \frac{|\mathcal{S}|(1-\gamma)^{-2}}{\eta_T^v} + \frac{1}{2}\sum_{t=1}^T \eta_t^v \|\tilde{\boldsymbol{g}}^{\boldsymbol{v}}(\boldsymbol{v}_t, \boldsymbol{\mu}_t)\|_2^2$$

$$= \sqrt{2}\cdot\sqrt{|\mathcal{S}|}(1-\gamma)^{-1}\cdot\sqrt{\sum_{t=1}^T \|\tilde{\boldsymbol{g}}^{\boldsymbol{v}}(\boldsymbol{v}_t, \boldsymbol{\mu}_t)\|_2^2}$$

$$+ \frac{\sqrt{2}}{4}\cdot\sqrt{|\mathcal{S}|}(1-\gamma)^{-1}\cdot\sum_{t=1}^T \frac{\|\tilde{\boldsymbol{g}}^{\boldsymbol{v}}(\boldsymbol{v}_t, \boldsymbol{\mu}_t)\|_2^2}{\sqrt{\sum_{i=1}^t \|\tilde{\boldsymbol{g}}^{\boldsymbol{v}}(\boldsymbol{v}_i, \boldsymbol{\mu}_i)\|_2^2}}$$

$$\leq \frac{3}{2}\sqrt{2}\cdot\sqrt{|\mathcal{S}|}(1-\gamma)^{-1}\cdot\sqrt{\sum_{t=1}^T \|\tilde{\boldsymbol{g}}^{\boldsymbol{v}}(\boldsymbol{v}_t, \boldsymbol{\mu}_t)\|_2^2},$$

where the second equality comes from the fact that $\tilde{\boldsymbol{g}}^{\boldsymbol{v}}$ only depends on $\boldsymbol{\mu}$ based on its construction rule. The first inequality comes from Theorem 6.10 in (Orabona, 2019). The third equality comes from the definition of $\eta_t^v$. The second inequality comes from Lemma 4.13 in (Orabona, 2019). Therefore,

$$\mathbb{E}\left[\sum_{t=1}^T \hat{f}^{(\boldsymbol{v})}(\boldsymbol{v}_t, \boldsymbol{\mu}_t) - \sum_{t=1}^T \hat{f}^{(\boldsymbol{v})}(\boldsymbol{v}, \boldsymbol{\mu}_t)\right] \leq \frac{3}{2}\sqrt{2}\cdot\sqrt{|\mathcal{S}|}(1-\gamma)^{-1}\cdot\sqrt{\sum_{t=1}^T \mathbb{E}\left[\|\tilde{\boldsymbol{g}}^{\boldsymbol{v}}(\boldsymbol{v}_t, \boldsymbol{\mu}_t)\|_2^2\right]}$$

$$\leq 3\sqrt{|\mathcal{S}|}(1-\gamma)^{-1}\sqrt{T}, \tag{21}$$

where the first inequality comes from Jensen Inequality. The second inequality comes from Lemma C.1. Consequently, combine (21), (20) into (18), we finish the proof.

### C.3. Proof for Theorem 4.5

The proof of theorem requires the following Lemma from (Jin & Sidford, 2020), which states that an expected $\epsilon$-optimal solution for the minimax problem (4) implies an expected $O((1-\gamma)^{-1}\epsilon)$-optimal policy:

**Lemma C.3.** *(Lemma 9, (Jin & Sidford, 2020)) Given an expected $\epsilon$-optimal solution for minimax problem (4) with initial distribution $\boldsymbol{q}$, dubbed as $(\boldsymbol{v}^{(\epsilon)}, \boldsymbol{\mu}^{(\epsilon)})$, let $\pi^{(\epsilon)}$ satisfies $\mu_{(i,a)}^{(\epsilon)} = \lambda_i \pi_{(i,a)}^{(\epsilon)}$ for some $\boldsymbol{\lambda} \in \Delta^{|\mathcal{S}|}$ and $\pi_i^{(\epsilon)} \in \Delta^{|\mathcal{A}_i|}, \forall i \in \mathcal{S}$. Then $v^*(\boldsymbol{q}) \leq \mathbb{E}[v^{\boldsymbol{\pi}^{(\epsilon)}}(\boldsymbol{q})] + 3(1-\gamma)^{-1}\epsilon.$*

By Lemma C.3, an expected $(1-\gamma)\epsilon/3$-optimal solution for minimax problem (4) implies an expected $\epsilon$-optimal policy. Choose an appropriate optimization length $T$ such that $\text{Err}_v$, $\text{Err}_{\mu,1}$, $\text{Err}_{\mu,2}$ satisfies $\leq (1-\gamma)\epsilon/9$. Let the required solution be $T_v'$, $T_{\mu,1}'$, $T_{\mu,2}'$, respectively. For any optimization length $T \geq \max\{T_v', T_{\mu,1}', T_{\mu,2}'\}$, the algorithm outputs an expected $\epsilon$-optimal policy. Additionally, each step in the optimization process requires exactly 2 samples of state transitions (construct $\tilde{\boldsymbol{g}}_t^{\boldsymbol{v}}, \tilde{\boldsymbol{g}}_t^{\boldsymbol{\mu}}$). Altogether, the Theorem is established.

## D. Numerical Experiments

In this section, we provide numerical experiments to validate our approach. We simulate a MDP instance with total 31 states.

We consider the MDP instance defined as follows. There are total 31 states and $\mathcal{S} = \{0, 1, ..., 30\}$. For each state $s$ such that $s \geq 1$, it can choose two actions: "stay" and "leave". If it chooses "stay", then it receives reward 0.001 and stays in state $s$ with probability 1. If it chooses "leave", it receives reward 0.5. Then, it stays in state $s$ with probability 0.3 and transits to state 0 with probability 0.7. For state 0, it can also choose two actions: "left", "right". For both actions, it receives reward 1 with certainty. If it chooses "left", then it transits to state $s$ for each $s \geq 1$ with probability $1/30$. If it chooses "right", then it stays in state 0 with probability 0.5 and transits to state $s$ for each $s \geq 1$ with probability $1/60$. It is straightforward to see that the optimal policy for each state $s$ that $s \geq 1$ is "leave" and for state 0 is "right", since the action that possibly results in arriving and staying in state 0 will always bring in higher reward.

We set $\gamma = 0.5$, $\epsilon = 0.05$, $\boldsymbol{q} = (1/31, 1/31, \cdots, 1/31)$. We vary the number of samples $T$ from 1000 to 39000. When applying OpPMD-NAC, we set the prediction matrix as follows: Denote action "stay" ("left") as $a_1$, "leave" ("right") as $a_2$, for each $i, j \in \mathcal{S}$, $a \in \{a_1, a_2\}$,

$$\hat{p}(j|i, a) = 1/31.$$

We evaluate and compare three algorithms in this environment: (1) OpPMD-AC: Algorithm 1 with accurate prediction input, i.e. $\hat{P} = P$. (2) OpPMD-NAC: Algorithm 1 with inaccurate prediction, i.e. $\hat{P} \neq P$ and $\text{Dist}(\hat{P}, P) > 1$. (3) SMD-DMDP-JINSID: Algorithm 1 in (Jin & Sidford, 2020), designed to solve DMDPs without any predictions.

The following results present the value function (for state 0) derived from the optimal policy produced by each algorithm. We emphasize that the strong performance of OpPMD-AC does not imply that it is uniformly superior to standard DMDP methods like (Jin & Sidford, 2020; Sidford et al., 2018; Wainwright, 2019). Instead, its advantage comes from utilizing accurate transition predictions, which the others lack. These findings illustrate not only the practical benefit of having access to reliable prior information, but also underscore the importance of an algorithmic design (like ours), that can effectively utilize such information. We believe that these results provide empirical validation to back up our claims.

| # samples | OpPMD-AC | OpPMD-NAC | SMD-DMDP-JINSID |
|---|---|---|---|
| 1000 | 0.988382 | 0.934327 | 0.787048 |
| 3000 | 1.115354 | 0.915718 | 0.808952 |
| 5000 | 1.210028 | 1.080434 | 0.835617 |
| 7000 | 1.209709 | 1.102027 | 0.858689 |
| 9000 | 1.240455 | 1.128366 | 0.886931 |
| 11000 | 1.264326 | 1.186346 | 0.913416 |
| 13000 | 1.253440 | 1.177276 | 0.938176 |
| 16000 | 1.258570 | 1.209913 | 0.965447 |
| 19000 | 1.260533 | 1.198441 | 0.988704 |
| 22000 | 1.278497 | 1.219869 | 1.013929 |
| 25000 | 1.273420 | 1.225140 | 1.037900 |
| 28000 | 1.253971 | 1.224485 | 1.050103 |
| 31000 | 1.261562 | 1.222145 | 1.073147 |
| 35000 | 1.277084 | 1.218521 | 1.090658 |
| 39000 | 1.270607 | 1.235767 | 1.108360 |

*Table 1.* Performance comparison

