# OpenReview forum: "Efficiently Solving Discounted MDPs via Predictions with Unknown Prediction Errors"
_ICML.cc/2026/Conference — ICML 2026 regular_

### Official Review · Reviewer_nD49 · 2026-02-27

**Soundness:** 3
**Presentation:** 2
**Significance:** 2
**Originality:** 1
**Overall Recommendation:** 2
**Confidence:** 5

**Summary:**

This paper focuses on solving Markov decision processes (MDPs) with a generative model. The main novelty is to consider the case where we are given a prediction of the unknown transition probabilities, without being given its error (distance to the true nominal parameters). The authors design a predictive ("optimistic") primal-dual algorithm based on the LP formulation of MDPs, and derive a minimax lower bound on the sampling complexity for this problem.

**Compliance With Llm Reviewing Policy:**

Affirmed.

**Final Justification:**

I have the two responses that the authors provided to my original comments and my second assessment. I remain unconvinced of the strength of the methodological contributions (that is, the paper looks incremental to me), and I keep my score.

**Key Questions For Authors:**

Theorem 4.5 + l347 :  I don’t see how your bound improves upon Jin and Sidford (2020)’s results of $\tilde{O}((1-\gamma)^{-4} N \espilon^{-2})$. If $Dist  \geq 1$ that is not the case, right ? l356, you should be more specific about what happens when $Dist = \Omega(1)$, what does ``gracefully defaults” mean? What is the standard performance?

**Limitations:**

Yes.

**Strengths And Weaknesses:**

**Soundness:** The algebra in the paper appears to be correct.


 **Presentation:** I don't have much to say about the presentation. The paper could be more polished, e.g. for the second paragraph of the introduction: usually the notations for the models are introduced later in the paper, not in the introductio; and In-line citations do not need parenthesis, e.g. l219 should be: Puterman, 2014 shows …

**Significance and originality:** This is the main drawback of this work. The results appear very incremental. The results in Section 3 are extensions of the analysis of  Gheshlaghi Azar et al., 2013, while Section 4.1 is an extension of the results by Sidford and Jin (2020). The analysis of the main theorme (Theorem 4.5) appears to follow from standard analysis of optimistic online mirror descent, as highlighted by the numerous references to the classical textbook of Francesco Orabona in Appendix C.2. I don't think that this paper passes the bar for ICML.

---

> ### Author Rebuttal · Authors · 2026-03-27
>
> Thank you for the review!
>
> **On Presentation** We acknowledge that certain writing aspects, such as citation formatting and the placement of notation, can be improved, and we will revise accordingly. However, we believe that the current presentation is sufficiently clear for understanding the key contributions and technical arguments of our work.
>
> **On Incremental results** We respectfully disagree that our results are incremental. We agree that our work builds on existing tools. However, the problem we address and our use of these tools are both novel. Two closely related papers study learning MDPs with predictions. Feng et al., (2019) studies DMDPs with an approximate model, but assumes knowledge of the TV distance between it and the true model. Golowich & Moitra, (2022) considers online tabular MDPs with predictions on Q-values, but requires knowledge of the prediction error to achieve the best possible bound. We find this assumption unrealistic: in practice, it is almost always hard to obtain reliable knowledge of the prediction error.
>
> To overcome this limitation, we identify that tools from optimistic mirror descent and parameter-free algorithm can help to handle the challenge of incorporating black-box predictions of unknown accuracy under the special structure of MDP. However, this is far from straightforward and requires several novel analytical components.
>
> Specifically, our algorithm incorporates two types of gradients: standard stochastic gradients constructed from sampling, and new predicted gradients constructed using the prediction matrix. This structure introduces non-trivial interactions, and thus new techniques are required. We apply the subadditivity of infinity norms to separate the two components. For predicted gradients, we bound their magnitude via the prediction error. For stochastic gradients, we introduce a variance reduction mechanism to control cumulative norms over time. Another challenges lies in ensuring parameter-free adaptivity. We leverage a discrete-to-integral bounding argument for nonincreasing functions to design adaptive learning rates that automatically adapt to the unknown prediction quality, entirely eliminating the need for this knowledge.
>
> Putting these together, we obtain a best-of-both-worlds algorithm that: (a) significantly improves upon Jin and Sidford (2020), the primal-dual state-of-the-art performance, when the prediction is accurate, and (b) gracefully recovers the guarantee of Jin and Sidford (2020) when the prediction is poor. We achieve both without knowing which case holds. We view our work as an important advance at the intersection of MDP, optimization, and algorithms with predictions.
>
> We also respectfully disagree that our results in Section 3 are extensions of Gheshlaghi Azar et al., (2013). While there are surface similarities, the core problem are fundamentally different. Gheshlaghi Azar et al., (2013) studies the classical minimax bound for computing an $\epsilon$-optimal policy in DMDPs with a generative model, without any additional information. Our setting introduces a prediction matrix of unknown quality, raising a new challenge: the algorithm must simultaneously solve the MDP and determine whether the prediction is helpful. Our result refutes the possibility of simultaneously testing whether the prediction is close to the true P and learning the MDP with fewer samples than the minimax bound in the worst case.
>
> **On Clarification on Theorem 4.5** We acknowledge that the presentation could be clearer on this point and provide the following clarification. By the definition of Dist, we have
>
> $
> \text{Dist}(\boldsymbol{\text{P}},\hat{\boldsymbol{\text{P}}}) = \max_{(i,a) \in \mathcal{N}}  \sum_{j \in \mathcal{S}} \left | \hat{p}(j\mid i,a)  - p(j \mid i,a)  \right| \le \max_{(i,a) \in \mathcal{N}}  \sum_{j \in \mathcal{S}} ( \hat{p}(j\mid i,a)  + p(j \mid i,a)  ) = \max_{(i,a) \in \mathcal{N}} 2 = 2.
> $
>
> So we always have $Dist = O(1)$. Therefore, the bound in Theorem 4.5, $\tilde{O}  ( \max  (\{T_{v},T_{\mu,1},T_{\mu,2}  \} ))$,  is **uniformly** better than $\tilde{O}((1-\gamma)^{-4}N\epsilon^{-2})$ from Jin and Sidford (2020), since each of $T_{v}$, $T_{\mu,1}$, and $T_{\mu,2}$ is smaller than $(1-\gamma)^{-4}N\epsilon^{-2}$ by the expression in Theorem 4.5.
>
> To clarify "it gracefully defaults to the standard performance": since Dist is always bounded by 2, even for an
> arbitrarily poor prediction, our algorithm can recover the performance of the primal-dual state-of-the-art algorithm Jin and Sidford (2020). Furthermore, when the prediction is sufficiently (e.g. $\text{Dist}^2 \le O((1-\gamma)^2\epsilon)$ and $|\mathcal{A}| > \Omega((1-\gamma)^{-1})$), our bound can even improve upon the minimax sample complexity bound $\tilde{O}((1-\gamma)^{-3}N\epsilon^{-2})$.

---

> > ### Author Rebuttal · Reviewer_nD49 · 2026-04-02
> >
> > I would like to thank the authors for taking the time to respond to my comments and for clarifying the extent of their main contributions and the main difference with existing work. After reading the responses, my evaluation of the paper did not change. The paper reads as a combination of several techniques, but I don’t see a major methodological or practical advancements that would be worth publishing at a top ML conference like ICML. For all these reasons I keep my evaluation and encourage the authors to resubmit to another conference after expanding their results.

---

> > > ### Author Response · Authors · 2026-04-03
> > >
> > > Thank you for your response. We respect your assessment, but we believe it does not fully reflect the contribution of the paper.
> > >
> > > We would like to further emphasize that achieving best-of-both-worlds results without knowing the prediction quality is a major conceptual novelty, which is only possible via the novel technical considerations we introduce. Reducing this to "a combination of several techniques" overlooks the core contribution of the work.
> > >
> > > Previous works on stochastic optimization with possibly biased predictions, such as RL(DMDP) [1], RL(online tabular MDP) [2], MAB [3], newsvendor and pricing problems [4], all crucially require knowing the prediction quality, via an upper bound on the prediction error. Our work presents a drastic shift from this requirement by eliminating it entirely.
> > >
> > > We believe this is an important message for the research community: Incorporating predictions into algorithms has become increasingly popular, yet the need to know the prediction error remains a key bottleneck for practical applications. Our work directly addresses this bottleneck.
> > >
> > > [1] Feng, F., Yin, W., & Yang, L. F. (2019). How Does an Approximate Model Help in Reinforcement Learning?. arXiv preprint arXiv:1912.02986.
> > >
> > > [2] Golowich, N., & Moitra, A. (2022, June). Can Q-learning be improved with advice?. In Conference on Learning Theory (pp. 4548-4619). PMLR.
> > >
> > > [3] Cheung, Wang Chi, and Lixing Lyu. "Leveraging (biased) information: Multi-armed bandits with offline data." Forty-first International Conference on Machine Learning. 2024.
> > >
> > > [4] Besbes, Omar, Will Ma, and Omar Mouchtaki. "Beyond IID: Data-Driven Decision Making in Heterogeneous Environments." Management Science 71.12 (2025): 10538-10555.

---

### Official Review · Reviewer_f2Rr · 2026-03-07

**Soundness:** 3
**Presentation:** 3
**Significance:** 3
**Originality:** 3
**Overall Recommendation:** 4
**Confidence:** 3

**Summary:**

The paper studies infinite-horizon discounted Markov decision processes (DMDPs) under a generative model. Motivated by the "Algorithms with Advice" framework, the paper explores how black-box predictions of the transition matrix can improve sample efficiency.

First, the paper presents an impossibility result: without prior knowledge of the prediction accuracy, no sampling policy can achieve a sample complexity better than $\tilde{O}((1-\gamma)^{-3} N \epsilon^{-2})$ across all DMDP instances. To address this, the paper proposes Optimistic-Predict Mirror Descent (OpPMD), a primal-dual algorithm. OpPMD adaptively integrates the predicted transition matrix into the estimation of future gradients. By utilizing parameter-free, adaptive learning rates, the algorithm leverages predictions without needing to know the prediction error. Ultimately, OpPMD achieves a sample complexity that is uniformly better than the previous primal-dual SoTA bound.

**Compliance With Llm Reviewing Policy:**

Affirmed.

**Final Justification:**

My questions are resolved. I believe my current score accurately reflects the merits of the paper and I keep my positive score.

**Key Questions For Authors:**

**Q1**
In the mathematical proof for the impossibility result (Theorem 3.2), the condition $\gamma\geq 1/3$ is explicitly required. Is this a significant theoretical barrier? Anything special about this requirement?

**Q2**
If possible, it would be good to do an empirical verification on some very simple synthetic tabular MDPs to demonstrate the actual rate of acceleration when the prediction matrix is highly accurate compared to a baseline where it is uninformative or poorly estimated?

**Q3**
The paper mentioned function approximation as a future direction. What are the primary technical challenges in adapting this prediction-based optimistic mirror descent framework from tabular MDP to models like linear MDP?

**Limitations:**

yes

**Strengths And Weaknesses:**

**Soundness**

The paper is rigorous in its theoretical foundations, successfully framing the problem as a minimax bilinear optimization task. The impossibility result (Theorem 3.2) is technically sound, utilizing well-established divergence decomposition and change-of-measure arguments to prove the lower bound.

Weakness: There lacks empirical study. I understand this is a theoretical work. But some synthetic experiments or numerical simulation would be good to have.

**Presentation**

The presentation is good. The fundamental limitations (i.e. the impossibility result) from the proposed OpPMD is presented clearly. The mathematical definitions and notations are clear.


**Significance**

The improvement of the sample complexity bound for primal-dual methods is a good theoretical contribution for the RL and online learning field. The ability for an algorithm to safely default to standard performance when predictions are bad—without knowing the quality of the prior information—tackles a quite practical bottleneck in algorithmic design.
The minor weakness is that it currently only applies to tabular DMDP.


**Originality**

The paper introduces the idea of black-box transition matrix predictions under the context of ‘solving MDPs with a generative model’. Furthermore, using adaptive learning rates borrowed from deterministic optimization to entirely bypass the need for a known upper bound on the prediction error is a good idea.

---

> ### Author Rebuttal · Authors · 2026-03-26
>
> Thank you for your feedback! Now we address your concerns and questions.
>
> **Q1 on impossibility result** The condition $\gamma \ge 1 / 3$ is not a fundamental barrier. To obtain a meaningful lower bound, it suffices to require $\gamma \ge c$ for some constant $c$ that $c > 0$. In the current Theorem 3.2, we choose $c=1/3$ for convenience. If a smaller $c$ is chosen, the result still holds with a smaller absolute constant (smaller than current $1/300$) in the bound. This condition is merely a technical convenience rather than a fundamental restriction.
>
> **Q2 on numerical validations** We have conducted additional numerical experiments on synthetic tabular MDPs to verify our theoretical findings. The results are shown below. Please refer to our rebuttal response to Reviewer giwp for detailed experiment settings.
>
> We evaluate and compare three algorithms in this environment: (1) **OpPMD-AC**: our algorithm with accurate prediction input, i.e. $\hat{P} = P$. (2) **OpPMD-NAC**: our algorithm with inaccurate prediction, i.e. $\hat{P} \ne P$ and $\mathrm{Dist}(\hat{P},P) > 1$.  (3) **SMD-DMDP-JINSID**: Algorithm 1 in [1], designed to solve DMDPs without any predictions.
>
> | # samples | 1000 | 3000 | 5000 | 7000 | 9000 | 11000 | 13000 | 16000 | 19000 | 22000 | 25000 | 28000 | 31000 | 35000 | 39000 |
> |---|---|---|---|---|---|---|---|---|---|---|---|---|---|---|---|
> | **OpPMD-AC** | 0.988382 | 1.115354 | 1.210028 | 1.210970 | 1.240455 | 1.264326 | 1.253440 | 1.258570 | 1.260533 | 1.278497 | 1.273420 | 1.253971 | 1.261562 | 1.277084 | 1.270607 |
> | **OpPMD-NAC** | 0.934327 | 0.915718 | 1.080434 | 1.102027 | 1.128366 | 1.186346 | 1.177276 | 1.209913 | 1.198441 | 1.219869 | 1.225140 | 1.224485 | 1.222145 | 1.218521 | 1.235767 |
> | **SMD-DMDP-JINSID** | 0.787048 | 0.808952 | 0.835617 | 0.858689 | 0.886931 | 0.913416 | 0.938176 | 0.965447 | 0.988704 | 1.013929 | 1.037900 | 1.050103 | 1.073147 | 1.090658 | 1.108360 |
>
> The results illustrate both the practical benefit of having access to reliable prior information and the importance of an algorithmic design, like ours, that can effectively leverage such information without the need to know its quality.
>
> **Q3 on future direction** Extending our framework to linear MDPs or other function approximation settings would require a new optimization formulation with different objective functions and a correspondingly different gradient structure. Therefore, the way predictions are incorporated may need to be fundamentally redesigned. Even so, we believe that the gradient-based nature of our framework can adapt to such settings, and we view this as a promising and important future direction.
>
> [1] Jin, Y., & Sidford, A.(2020). Efficiently solving MDPs with stochastic mirror descent. In International Conference on Machine Learning (pp. 4890-4900). PMLR.

---

> > ### Author Rebuttal · Reviewer_f2Rr · 2026-04-03
> >
> > I thank the authors for their rebuttal. The additional numerical results are appreciated and I think they in general corroborated the theoretical claim.
> >
> > My questions are resolved. I actually believe my current score accurately reflects the merits of the paper and I keep my positive score.

---

> > > ### Author Response · Authors · 2026-04-07
> > >
> > > Thank you again for your valuable comment and positive review! The review has helped us identify the areas for improvement in our manuscript.

---

### Official Review · Reviewer_9Y5b · 2026-03-13

**Soundness:** 3
**Presentation:** 3
**Significance:** 2
**Originality:** 3
**Overall Recommendation:** 4
**Confidence:** 3

**Summary:**

This paper studies discounted infinite-horizon MDPs with a generative model, where the learner is also given a black-box prediction of the transition matrix, but does not know how acccurate it is. The question is whether such predictions can improve sample complexity while remaining robust when the prediction is poor. The paper claims that in the worst case, predictions with unknown error cannot beat the classic minimax sample complexity. The authors propose OpPMD that can exploit good predictions and improve over prior primal-dual methods without requiring knowledge of the prediction error.

**Compliance With Llm Reviewing Policy:**

Affirmed.

**Final Justification:**

During the rebuttal period, the authors answered my questions and addressed most of my concerns. Therefore, I keep my positive rating for this paper as weak accept.

**Key Questions For Authors:**

1. Is your upper bound tight as a function of Dist($P, \hat{P}$), or is there still room for improvement?
2. Is the black-box prediction $\hat{P}$ allowed to be arbitrarily biased or perhaps adversarial?
3. Can you give intuition for the specific design of the adaptive learning rates and why they eliminate the need to know the prediction error?
4. Is it possible to develop similar results with online access instead of random access?

**Limitations:**

yes

**Strengths And Weaknesses:**

Strengths:
- The problem is clean and well motivated.
- The impossibility result is conceptually important. It clarifies that predictions with unknown quality cannot improve worst-case minimax complexity.
- The paper is technically solid to the best of my knowledge, and positions itself well against prior primal-dual results.

Weaknesses:
- The improvement is mainly over primal-dual methods, not over the full minimax frontier in the worst case.
- The contribution is algorithmically interesting but is somewhat incremental if viewing it as brining prediction with unknown error into an existing mirror-descent framework.
- The results are confined to tabular discounted MDPs with a generative model, which is a fundamental but fairly idealized setting.

---

> ### Author Rebuttal · Authors · 2026-03-26
>
> Thank you for your review!
>
> **Q1** By Theorem 3.2, even with predictions, no algorithm can compute an $\epsilon$-optimal policy with fewer than $\tilde{o}((1- \gamma)^{-3}N\epsilon^{-2})$ samples for all DMDP instances without knowledge of the prediction error. This implies that no algorithm leveraging predictions can outperform the minimax sample complexity bound in the worst case, suggesting there is no room for improvement in terms of Dist.
>
> However, if the error Dist is known in advance, it becomes possible to derive a matching lower bound to establish the tightness of the Dist-dependence of our upper bound. We also believe that an algorithm with an improved sample complexity bound (given known Dist) is possible. A related consideration appears in [1], though without a matching lower bound to verify optimality. Investigating whether we can further improve is an interesting future direction.
>
> **Q2** Yes. The black-box prediction $\hat{P}$ is allowed to be arbitrarily biased or adversarial. When the prediction $\hat{P}$ is arbitrarily bad, our algorithm performance gracefully defaults to the performance of the best known primal-dual-type methods. When the prediction is sufficiently accurate, our algorithm can significantly improve upon that baseline. This demonstrates both the robustness and adaptivity of our algorithm OpPMD.
>
> **Q3** The key insight comes from the power of the optimistic mirror descent framework. Our algorithm first translates the prediction into "predicted gradients" (See Section 4.1.2), and the optimistic mirror descent framework enables us to incorporate these predicted gradients and benefit from their potential accuracy. When predictions are accurate, the predicted gradients are close to the true gradients, leading to faster convergence. Otherwise, the framework still recovers the worst-case guarantee.
>
> However, in the standard framework, achieving the best possible performance still requires the knowledge of desired accuracy level $\epsilon$ and the prediction error Dist to tune the optimal learning rate. To resolve this, we design adaptive learning rates, inspired by parameter-free algorithm design, that automatically adapt to unknown parameters. Putting these together, our algorithm entirely removes the need to know the prediction error in advance.
>
> **Q4, W3**  We interpret "online access" as referring to the setting where samples are collected sequentially rather than via a generative model. We believe extending our results to this setting is feasible and is an interesting future direction. More broadly, while our current framework focuses on tabular DMDPs under a generative model, the gradient-based structure of our framework is flexible and well-suited to incorporating predictions from various sources and to more complex RL settings (e.g., linear MDPs). We view our problem as a first step toward this direction, and we are interested in applying the idea of leveraging prediction in other more complex RL environments.
>
> **W1** We focus on primal-dual methods for two reasons. First, primal-dual methods are computationally efficient. Second, the optimistic mirror descent framework provides a way to incorporate predictions without requiring knowledge of the error, which is the key challenge we address. Extending this to improve upon the full minimax frontier in the worst case is an important open problem, and we view it as a crucial next step for future work.
>
> **W2** While we build on the mirror descent framework, the problem we study and our use of these tools are both novel. Two closely related papers study learning MDPs with predictions. [1] considers DMDPs with an approximate model but requires knowledge of the TV distance between the approximate and true model. [2] considers online tabular MDPs with predictions on Q-values but similarly requires knowledge of the prediction error to achieve the best bound. We find this assumption unrealistic. Such knowledge is almost never available in practice. We identify that optimistic mirror descent and parameter-free algorithm design can overcome this limitation. By carefully combining these ideas, we design a best-of-both-worlds algorithm that (a) significantly improves upon primal-dual state-of-the-art [3] when predictions are good, and (b) recovers the guarantee of [3] when predictions are arbitrarily poor. We achieve both without knowing which case we are in. We view this as an important advance at the intersection of MDP, optimization, and algorithms with predictions.
>
> [1] Feng, F., Yin, W., & Yang, L. F. (2019). How Does an Approximate Model Help in Reinforcement Learning?. arXiv preprint arXiv:1912.02986.
>
> [2] Golowich, N., & Moitra, A. (2022, June). Can Q-learning be improved with advice?. In Conference on Learning Theory (pp. 4548-4619). PMLR.
>
> [3] Jin, Y., & Sidford, A.(2020). Efficiently solving MDPs with stochastic mirror descent. In International Conference on Machine Learning (pp. 4890-4900). PMLR.

---

> > ### Author Rebuttal · Reviewer_9Y5b · 2026-04-03
> >
> > I thank the authors for the response. Most of my questions are resolved, and I keep my positive score.

---

> > > ### Author Response · Authors · 2026-04-07
> > >
> > > Thank you again for your thorough and insightful comment. We appreciate the time and effort you have dedicated to evaluating our work and providing constructive feedback.

---

### Official Review · Reviewer_giwp · 2026-03-14

**Soundness:** 3
**Presentation:** 3
**Significance:** 3
**Originality:** 3
**Overall Recommendation:** 4
**Confidence:** 3

**Summary:**

This paper studies the problem of prediction-augmented reinforcement learning for infinite-horizon discounted MDPs with a generative model. The setting assumes that the learner is given a black-box prediction of the transition matrix whose accuracy is unknown. The goal is to design algorithms that improve sample complexity when the prediction is accurate while remaining robust when it is inaccurate. The paper first establishes an impossibility result showing that if the prediction quality is completely unknown, no algorithm can uniformly improve the minimax sample complexity beyond the classical bound for solving discounted MDPs. The authors propose OpPMD which is a primal dual algorithm that integrates predicted transition information through optimistic gradient estimates. The paper provides a sample complexity bound that depends on the distance between the true and predicted transition kernel.

**Compliance With Llm Reviewing Policy:**

Affirmed.

**Key Questions For Authors:**

none

**Limitations:**

yes

**Strengths And Weaknesses:**

The paper studies how to safely leverage external predictions in reinforcement learning without knowing their accuracy. The formulation is natural and connects well with recent literature on prediction-augmented learning. The paper makes two contributions. First, it provides an impossibility result showing that prediction quality must influence achievable improvements. Second, they propose OpPMD algorithm analyze sample complexity bounds that smoothly interpolates between the classical minimax rate and improved rates when the prediction error is small. The contributions appear solid and help improve the understanding on use of synthetic data in learning.

Weakness: My main concern with the paper is that it is purely theoretical and does not provide any empirical evidence to back up the claims,

---

> ### Author Rebuttal · Authors · 2026-03-26
>
> Thank you for your response! We sincerely appreciate your thoughtful and positive feedback. Now we address your concern on **empirical validations**.
>
> We provide numerical experiments to validate our approach. We consider the MDP instance defined as follows. There are total $31$ states and $\mathcal{S} = 0,1,\ldots,30$. For each state $s$ such that $s \ge 1$, it can choose two actions: ''stay'' and ''leave''. If it chooses ''stay'', then it receives reward $0.001$ and stays in state $s$ with probability $1$. If it chooses ''leave'', it receives reward $0.5$. Then, it stays in state $s$ with probability $0.3$ and transits to state $0$ with probability $0.7$. For state $0$, it can also choose two actions: ''left'', ''right''. For both actions, it receives reward $1$ with certainty. If it chooses ''left'', then it transits to state $s$ for each $s \ge 1$ with probability $1/30$. If it chooses ''right'', then it stays in state $0$ with probability $0.5$ and transits to state $s$ for each $s \ge 1$ with probability $1/60$. It is straightforward to see that the optimal policy for each state $s$ that $s \ge 1$ is ``leave'' and for state $0$ is ''right'', since the action that possibly results in arriving and staying in state $0$ will always bring in higher reward.
>
> We evaluate and compare three algorithms in this environment: (1) **OpPMD-AC**: our algorithm with accurate prediction input, i.e. $\hat{P} = P$. (2) **OpPMD-NAC**: our algorithm with inaccurate prediction, i.e. $\hat{P} \ne P$ and $\mathrm{Dist}(\hat{P},P) > 1$. Here $\hat{P}$ is defined as follows. Denote action ''stay'' (''left'') as $a_1$, ''leave'' (''right'') as $a_2$, for each $i,j \in \mathcal{S}, a \in a_1,a_2$, $\hat{p}(j \mid i,a) = 1/31$. (3) **SMD-DMDP-JINSID**: Algorithm 1 in [1], designed to solve DMDPs without any predictions.
>
> The results below present the value function (for state 0) derived from the optimal policy produced by each algorithm. We emphasize that the strong performance of OpPMD-AC does not imply it is uniformly superior to standard DMDP methods like [1]. Instead, its advantage comes from utilizing accurate transition predictions, which the others lack. These findings illustrate not only the practical benefit of having access to reliable prior information, but also underscore the importance of an algorithmic design (like ours), that can effectively utilize such information. We believe these results provide empirical validation to back up our claims.
>
> | # samples | 1000 | 3000 | 5000 | 7000 | 9000 | 11000 | 13000 | 16000 | 19000 | 22000 | 25000 | 28000 | 31000 | 35000 | 39000 |
> |---|---|---|---|---|---|---|---|---|---|---|---|---|---|---|---|
> | **OpPMD-AC** | 0.988382 | 1.115354 | 1.210028 | 1.210970 | 1.240455 | 1.264326 | 1.253440 | 1.258570 | 1.260533 | 1.278497 | 1.273420 | 1.253971 | 1.261562 | 1.277084 | 1.270607 |
> | **OpPMD-NAC** | 0.934327 | 0.915718 | 1.080434 | 1.102027 | 1.128366 | 1.186346 | 1.177276 | 1.209913 | 1.198441 | 1.219869 | 1.225140 | 1.224485 | 1.222145 | 1.218521 | 1.235767 |
> | **SMD-DMDP-JINSID** | 0.787048 | 0.808952 | 0.835617 | 0.858689 | 0.886931 | 0.913416 | 0.938176 | 0.965447 | 0.988704 | 1.013929 | 1.037900 | 1.050103 | 1.073147 | 1.090658 | 1.108360 |
>
> [1] Jin, Y., & Sidford, A.(2020). Efficiently solving MDPs with stochastic mirror descent. In International Conference on Machine Learning (pp. 4890-4900). PMLR.

---

> > ### Author Rebuttal · Reviewer_giwp · 2026-04-01
> >
> > With the new example, the paper addresses my comment on limited experimental validation.

---

> > > ### Author Response · Authors · 2026-04-07
> > >
> > > Thank you again for your review, which we have carefully addressed in the rebuttal!

---

### Decision · Program_Chairs · 2026-04-30

**Decision:**

Accept (regular)

**Comment:**

This paper proposes an adaptive algorithm for solving MDPs with black-box transition predictions, crucially eliminating the need to know the prediction error a priori.

Strengths: Successfully removes a restrictive assumption in the RL literature. The work is technically rigorous.

Weaknesses: As argued by reviewer nD49, the core methodology relies heavily on standard parameter-free optimistic online learning techniques, making the mathematical contribution somewhat incremental.

Justification:
While the underlying online optimization tools are not novel, synthesizing them to solve a bottleneck in RL is a practically useful contribution. The paper represents a rigorous, incremental advance rather than a groundbreaking leap, making it a clear Weak Accept.